# Reproducible determination of dissolved organic matter photosensitivity

Alec W. Armstrong[1,2], Leanne Powers[1], Michael Gonsior[1]

[1]University of Maryland Center for Environmental Science, Chesapeake Biological Laboratory, Solomons, Maryland 20688
[2]Department of Entomology, University of Maryland, College Park, Maryland 20742

*Correspondence to*: Alec W. Armstrong (aarmstr1@umd.edu) and Michael Gonsior (gonsior@umces.edu)

**Abstract.** Dissolved organic matter (DOM) connects aquatic and terrestrial ecosystems, plays an important role in C and N cycles, and supports aquatic food webs. Understanding DOM chemical composition and reactivity is key to predict its ecological role, but characterization is difficult as natural DOM is comprised of a large but unknown number of distinct molecules. Photochemistry is one of the environmental processes responsible for changing the molecular composition of DOM and DOM composition also defines its susceptibility to photochemical alteration. Reliably differentiating the photosensitivity of DOM from different sources can improve our knowledge of how DOM composition is shaped by photochemical alteration and aid research into photochemistry's role in various DOM transformation processes. Here we describe an approach to measure and compare DOM photosensitivity consistently based on the kinetics of changes in DOM fluorescence during 20h photodegradation experiments. We identify several methodological choices that affect photosensitivity measurements and offer guidelines for adopting our methods, including use of a reference material, precise control of conditions affecting photon dose, leveraging actinometry to estimate photon dose instead of expressing results as a function of exposure time, and frequent (every 20 minutes) fluorescence and absorbance measurements during exposure to artificial sunlight. We then show that our approach can generate photosensitivity metrics across several sources of DOM, including freshwater wetlands, a stream, an estuary, and *Sargassum sp*. leachate, and observed differences in these metrics that may help identify or explain differences in their composition. Finally, we offer an example applying our approach to compare DOM photosensitivity in two adjacent freshwater wetlands as seasonal hydrologic changes alter their DOM sources.

# 1 Introduction

The photochemical reactivity of dissolved organic matter (DOM) is inherently linked to its composition and photochemical behavior reflects compositional differences between samples. Several authors have discussed the fundamental processes involved in light absorption by DOM and the phenomena that may follow (Miller, 1998; Sharpless et al., 2014) , including loss of absorbance (Del Vecchio and Blough, 2002), production of new substances (Gonsior et al., 2014; Blough and Zepp, 1995; Bushaw et al., 1996; Moran and Zepp, 1997), and loss of fluorescence (Blough and Del Vecchio, 2002). Absorption spectra and derived values such as spectral slopes and their ratios have long been used to characterize DOM (Blough and Del Vecchio, 2002; Helms et al., 2008; Twardowski et al., 2004). Fluorescence measurements arise from only a fraction of chromophoric DOM (CDOM) but are sensitive to small variations in DOM chemical composition (Blough and Del Vecchio, 2002). To the extent that photochemical reactivity is a property of DOM chemical composition (Boyle et al., 2009; Cory et al., 2014; Del Vecchio and Blough, 2004; Gonsior et al., 2013, 2009; Wünsch Urban J. et al., 2017), comparing the potential for photochemical transformation of different DOM sources or treatments (hereafter called photosensitivity) may be a useful tool in the continuing effort to characterize DOM composition and to describe its susceptibility to sunlight-induced degradation. Such comparisons require robust methods that are sensitive enough to discern ecologically and chemically relevant differences between distinct DOM sources.

Research across ecosystem settings has measured changes in optical properties following sunlight or simulated-sunlight irradiation to infer changes in DOM composition. A general discussion of this approach and its bases has been previously published (Hansen et al., 2016; Kujawinski et al., 2004; Sulzberger and Durisch-Kaiser, 2009). Examples of recent research using photochemical changes to make ecologically significant distinctions between DOM samples collected in specific ecosystems have been described in detail elsewhere (Gonsior et al., 2013; Laurion and Mladenov, 2013; McEnroe et al., 2013; Minor et al., 2007). DOM photodegradation itself has ecological consequences, affecting overall carbon (C) cycling (Anesio and Granéli, 2003; Obernosterer and Benner, 2004), microbial heterotrophy of DOM (Amado et al., 2015; Cory et al., 2014; Lapierre and del Giorgio, 2014), and algal and submerged plant primary productivity (Arrigo and Brown, 1996; Thrane et al., 2014).

Experimental approaches connecting DOM chemical composition, its optical properties and their photochemical bases, and relevant ecological phenomena typically expose natural DOM samples to natural or simulated sunlight and measure the change in optical properties over time. *In situ* experiments have been used to explore the role of photodegradation relative to other transformations of DOM in aquatic ecosystems but field studies are difficult if not impossible to reproduce (Cory et al., 2014; Groeneveld et al., 2016; Laurion and Mladenov, 2013). Laboratory-based irradiation experiments may allow greater reproducibility and logistical flexibility. Laboratory photodegradation experiments have tested the potential ecological significance of photodegradation and explored the fundamental photochemical mechanisms involved in photobleaching (Chen

and Jaffé, 2016; Del Vecchio and Blough, 2002; Goldstone et al., 2004; Hefner et al., 2006). These experiments usually involve simultaneous irradiation of DOM in several sample vials under polychromatic or monochromatic light. Vials are then destructively sampled for DOM measurements at intervals throughout the experiment, or simply compared before and after light exposure. While powerful, these experiments require a trade off in effort between reproducibility and temporal resolution.

Replicate vials are often sampled to ensure precision and improve reproducibility, but lamp space is finite, limiting temporal sampling resolution.

Continuous measurement of a single sample undergoing controlled photoirradiation offers an alternative experimental approach. The kinetics of DOM fluorescence loss during photoirradiation experiments have been recently described (Murphy

et al., 2018; Timko et al., 2015). These studies leveraged novel time series of frequent measurements (e.g. every 20 minutes) of fluorescence and UV-Vis absorption which allowed modeling of distinct reactive components. Fluorescence losses were best described by the sum of two exponential decay terms, allowing straightforward and precise modeling of photosensitive fluorescence signals that degraded quickly, which may reflect chemically distinct processes contributing to fluorescence loss during photodegradation. This approach may offer the resolution required to compare photosensitivity between samples with

small but ecologically significant differences in DOM composition

The goals of this study are to 1.) identify methodological barriers to reproducible determination of DOM photosensitivity and offer experimental guidelines to improve studies of DOM photodegradation kinetics; 2.) test our approach on samples from various environmental settings to see if our derived metrics of photosensitivity might respond to variability in DOM

composition; and 3.) analyze photosensitivity differences between different DOM sources in detail to better understand the links between DOM composition, environmental setting, and photochemical degradation processes. In a series of experiments, we explored potential sources of variability in photodegradation kinetics stemming from experimental conditions and methodology. We further develop a previously described experimental setup (Timko et al., 2015), showing results are reproducible under controlled conditions using a common reference material, and suggest a set of best practices for collecting

reproducible and high resolution time series of fluorescence measurements during experimental irradiation of a single sample. Then we apply this approach to several natural DOM sources by building on and exploring new dimensions of an established modeling framework (Murphy et al., 2018) to identify photosensitivity differences that may be ecologically relevant. Finally, we thoroughly test DOM from two wetlands to show how these differences in photosensitivity metrics may help us link DOM composition to ecological phenomena.

## 2 Materials and procedures

### 2.1 Photoirradiation system

We needed our system to irradiate samples without self-shading even at relatively high CDOM concentrations. The photoirradiation system circulates an aqueous sample between a mixing reservoir (i.e. equilibration flask), a solar simulator, and a spectrofluorometer, similar to a system described previously (Timko et al., 2015). A photograph of the system can be found in Appendix A (Fig. A1). Samples were continuously circulated between a central mixing reservoir and system components were connected by PEEK tubing (LEAP PAL Parts & Consumables, 0.0625" OD/0.030" ID). The central reservoir was a 25 mL borosilicate equilibrator flask with a magnetic stir bar constantly rotating at its bottom, at a speed low enough to prevent visible bubbles from forming. Sample gently dripping from flow lines into the equilibrator ensured sample remained oxygenated during photodegradation. A micro gear pump (HNP Mikrosysteme, mzr-4665) was used to pump the sample with almost pulse-less flow through the system at a rate of $1.5\pm0.1$ ml min$^{-1}$. The spectrophotometer flow cell and equilibrator flask were surrounded by a circulating water jacket set to 25 °C. To prevent contamination or the establishment of microbes that could degrade DOM during experiments, the system was flushed with 0.1 M NaOH between experiments then thoroughly flushed with ultrapure water. Ultrapure water for blanks was circulated for at least 10 minutes before checking absorbance and fluorescence for signs of contamination. If blank contamination persisted after subsequent rinses, the system was flushed with isopropanol and thoroughly rinsed with ultrapure water before checking for contamination by examining optics and testing [DOC].

Samples were irradiated as they were slowly pumped through a custom-built flow cell (SCHOTT Borofloat borosilicate glass, Hellma Analytics, 70 to 85% transmission between 300 and 350 nm, 85% transmission at wavelengths >350 nm), with a total exposure path area of 101 cm$^2$ arranged in an Archimedean spiral and returned to the equilibrator flask. This 20x20 cm borosilicate spiral flow cell had a 1 mm deep x 2 mm wide long flow path covering the irradiation area and was located underneath a solar simulator (Oriel Sol2A) with a 1,000 W Xe arc lamp equipped with an air mass (AM) 1.5 filter. Lamp output was checked periodically using an Oriel PV reference cell set to one sun which corresponds here to exactly 1,000 W m$^{-2}$ and lamp power was held constant during irradiation experiments using a Newport 68951 Digital Exposure Controller. Another tubing carried the sample from the equilibrator flask to a temperature-controlled square quartz fluorescence flow cell (1 cm x 1 cm) located within a Horiba Jobin Yvon Aqualog spectrofluorometer.

Total sample exposure varied depending on the total volume in the photodegradation system. We controlled volume by completely filling the tubing and flow cells (12.2 mL volume) and adjusting volume added to the equilibration flask. We used nitrite actinometry to calculate photon flux based on the response bandwidth between 330 and 380 nm of the nitrite actinometer (Jankowski et al., 1999, 2000). Briefly, a solution of 1 mM sodium nitrite, 1 mM benzoic acid, and 2.5 mM sodium bicarbonate was circulated through the irradiation system with regular measurements of fluorescence emission at 410 nm after

excitation at 305 nm. Results are compared against a fluorescence calibration curve using 0-5 μm salicylic acid fluorescence to calculate formation of salicylic acid from benzoic acid (mediated by hydroxyl radicals formed during photolysis of nitrite)

as a function of time. This is then used to calculate photon exposure as a function of time. Actinometer experiments were repeated with 0.5, 5, and 10 mL of actinometer solution added to the equilibration vessel after filling flow lines. Average July solar irradiance was modeled using the System for Transfer of Atmospheric Radiation model (Ruggaber et al., 1994) calculated just below the water surface as described previously (Fichot and Miller, 2010). With 10 mL volume added to the equilibrator (our typical experimental conditions), a 20 h irradiation experiment was equivalent to 1.0 day of exposure between

330 – 380 nm at 45 °N latitude in mid-July where one day is ~15.75 h long. For the lowest total volume used here (0.5 mL in the equilibrator, total volume 12.7 mL), photon dose was 1.7 times higher than this estimate. We calculated a mean photon flux of $3.9 \times 10^{-5}$ mol photons $m^{-2}s^{-1}$ for experiments with 10 mL sample added once flow lines were filled (total sample volume 22.2 mL), based on a mean photon exposure of 0.23 μmol photons $cm^{-2}$ $min^{-1}$ (5 trials, standard deviation 0.0045).

Past experiments revealed the importance of pH control on DOM fluorescence and photodegradation kinetics (Timko et al., 2015). We adjusted initial sample pH to 3.0 (+ 0.2) with HCl but did not control pH by autotitration. At pH 3 natural organic acids should generally be protonated regardless of compositional differences between DOM sources, which should prevent solution pH change due to the photoproduction of $CO_2$ (Ritchie and Perdue, 2003). Starting at pH 3 and equilibrating the sample in an air-filled reaction vessel ensured minimal pH change during irradiation, never changing by more than 0.2 pH

units, in line with expectations from work on mechanisms explaining pH decreases during photooxidation (Xie et al., 2004).

## 2.2 Optical measurements

We used a Horiba Jobin Yvon Aqualog spectrofluorometer to collect time series of UV-Vis absorbance and excitation-emission matrix (EEM) fluorescence spectra throughout experiments. UV-Vis absorbance was measured at 3 nm intervals between 600 and 230 nm. Fluorescence excitation occurred at the same intervals, and emission spectra were recorded from 600 to 230 nm

at 8 pixel CCD resolution, or approximately 3.24 nm intervals. EEMs integration times were 1 second. Milli-Q water (18.2 MΩ-cm) adjusted to pH 3 with concentrated HCl was circulated through the system and used as a measurement blank immediately prior to each experiment.

## 2.3 Experiments

Several sets of experiments explored method reproducibility, sensitivities to experimental conditions, and differences between

DOM sources. For our first goal of identifying methodological barriers to reproducible determination of DOM photosensitivity we varied the concentrations and volumes of SRNOM PPL extracts added to the photodegradation system to test their influence on degradation kinetics. Different researchers in our group then repeated experiments with SRNOM PPL extracts to test reproducibility. We explored effects of storage time on filtered water sample photodegradation results. We next compared SRNOM PPL extracts and SRNOM reference material reconstituted in ultrapure water (RO SRNOM) to test the effect of

extraction on photodegradation kinetics. We approached our second goal – demonstrating the utility of our approach as a measure of DOM photosensitivity – by applying methodological guidelines developed in our tests of SRNOM to PPL extracts of DOM from a variety of aquatic ecosystem settings and sources (see section 2.4). Finally, we ran experiments comparing photosensitivity of DOM sampled from two adjacent freshwater sites in different seasons to better understand the links between DOM composition, environmental setting, and photochemical degradation processes.


In each experiment, a sample was exposed to 20 hours of simulated sunlight, and EEM spectra were collected (using the "Sample Q" feature in Aqualog software) starting immediately before irradiation began with a 17.5 minute interval between each scan, generating a time series of 60 EEM spectra for each experiment. Where applicable, time of EEM collection was converted to cumulative photon exposure (mol photon m$^{-2}$) by multiplying time by calculated photon flux (mol photon m$^{-2}$ s$^{-1}$) using actinometry results generated with the same sample volume.


### 2.4 Sample materials

We used Suwannee River natural organic matter (SRNOM) obtained from the International Humic Substances Society as a reference material (catalog no. 2R101N, isolated by reverse osmosis; Green et al., 2014). Freeze-dried SRNOM was dissolved in Milli-Q water and was prepared less than one week prior to use (hereafter called RO SRNOM). Dilutions approximately

corresponded to a dissolved organic carbon (DOC) concentration of 5 mg C l$^{-1}$.This is well below the [DOC] range found in SRNOM source material before it was extracted, but within the range of other aquatic DOM sources dominated by terrestrially-derived DOM. Additionally, SRNOM solid phase extracts using the Agilent PPL resin were extracted in May 2012 during the same time the SRNOM standard material was isolated, and were prepared directly before irradiation experiments (see details below).


Additional water samples were collected across a variety of aquatic ecosystems to explore the range of our approach and to validate it. Sample sources include two freshwater wetland sites (Caroline County, Maryland, USA), one perennial stream (Parker's Creek, Calvert County, Maryland, USA, collected September 2017), one estuary (Delaware Bay, USA, collected July 2016), and leachate from live *Sargassum sp*. collected in Bermuda in July 2016 (Powers et al., 2019). These samples were

0.7 µM filtered within 24 hours of collection through combusted (500°C) Whatman GF/F filters and acidified to pH 2 using concentrated HCl (Sigma, 32% pure) before solid-phase extraction. The true pore size used in this pre-filter step was probably smaller than 0.7 µm (e.g. 0.3 µm in Nayar and Chou, 2003). All samples, whether whole water or solid-phase extracts redissolved in water, were filtered through syringe-mounted 0.2 µm cellulose acetate filters that were pre-rinsed with > 30 mL ultrapure C-free water.


Samples from the two freshwater wetland sites are used in the more detailed comparison presented in Section 3.3 and hence these sites merit additional description. Small topographic depressions are common throughout the interior of Delmarva

Peninsula. These depressions persist in this low-elevation, low-relief landscape, and regular seasonal inundation has led to the development of wetland soils and biota in many of these depressions. Depressions on land not drained for agriculture are inundated for several months most years. Some do not exchange water through surface flow with perennial stream networks, while others sustain downstream connections through temporary surface channels for several months in the wettest months of the year (typically late winter-spring). These two sites, referred to as "smaller wetland" and "larger wetland", are adjacent but lie within distinct topographic depressions. Their inundated areas expand and contract with water level fluctuations, and both may go entirely dry at the surface in the summer. If water levels are sufficiently high, their surface waters merge, and a temporary channel may fill and sustain export flow to the perennial stream network. One sampling site is within the smaller depression, which mostly lacks submerged and emergent vegetation and is hemmed closely by trees. The other site is within a larger depression, where surface water is more exposed to light and features a variety of herbaceous submerged and aquatic plants. Experiments were run with DOM from both sites, sampled on three dates (2017-10-05, 2017-12-20, 2018-04-01).

Except for RO SRNOM samples used to test the effect of solid phase extraction and wetland samples used for the storage time experiment described below, all samples were solid-phase extracted using a proprietary styrene divinyl benzene polymer resin (Agilent PPL Bond Elut) following a procedure described previously (Dittmar et al., 2008). PPL extracts were used because our goal is to develop a reproducible method to compare photochemical behavior of natural organic matter without the influence of the sample matrix. Extracts allow longer storage, isolate organic matter from potentially photosensitive matrices, and capture representative photosensitive organic matter fractions (Murphy et al., 2018). While filtration to 0.2 μm should remove most viable microbes, microbial degradation may still be possible in filtered water if ultra-small microorganisms are present (Brailsford et al., 2017; Luef et al., 2015). Extraction removes this possibility.

Immediately prior to each experiment, 0.5-5 ml of the extract was evaporated under high-purity $N_2$ gas, dissolved in 30 ml ultrapure C-free Milli-Q water, and diluted to similar CDOM absorbance values to minimize any potential inner filter effects on fluorescence degradation kinetics. Absorbance (A) at 300 nm was used as a benchmark for dilution instead of adjustments based on measured [DOC] because it could be done quickly on the equipment used for the photochemical experiments and allowed consistent correction of inner filtering effects. We adjusted all samples (except for those used in the storage time experiments described below) to a raw absorbance of 0.12 ($\pm$ 0.01), which translates to a Napierian absorption coefficient ($a$) of 27.6 $m^{-1}$. Delaware Bay samples were too dilute to generate sufficient volume to fill the photoirradiation system, so several sample extracts from throughout the depth profile of a single sample station were combined prior to evaporation.

### 2.5 Data analyses

Fluorescence EEM spectra were inner-filter corrected and had 1st order Rayleigh scatter removed by the built-in Aqualog software (based on Origin). Second order Rayleigh scatter was removed using an in-house Matlab toolbox following methods

previously described (Zepp et al., 2004) . EEM spectra were normalized by dividing fluorescence measurements by the area of the water Raman scatter peak of the water blanks. Data were processed in Matlab R2018a using an in-house toolbox and the drEEM toolbox (Murphy et al., 2013). Absorbance data were converted to absorption coefficients using Eq. 1:

$$a(\lambda) = 2.303A(\lambda)/l \tag{1}$$

where $a$ is the absorption coefficient at wavelength $\lambda$, A is raw absorbance at wavelength $\lambda$, and $l$ is path length in m, here 0.01 (Hu et al., 2002).

We fitted a 4-component parallel factor analysis (PARAFAC) model to data from 3 SRNOM PPL extract experiments (60 EEMs each, 180 EEMs total). PARAFAC models with 3, 4, and 5 components were fitted to the 3 SRNOM PPL extract

experiment EEMs. The 4-component model was chosen as it exhibited better component spectral characteristics than the others. Emission spectra from components matched the 4 components identified in similar experiments (Murphy et al., 2018). Split-half validation is often used to validate PARAFAC models fitted to data sets where each EEM represents a different DOM source but may not be appropriate for data sets where EEMs are not independent. Instead, 4-component models were fitted from each of the three SRNOM PPL extract experiments individually to confirm each experiment's data led to the same

PARAFAC model, then the model built from all three experiments was compared to each of these. All comparisons were confirmed using Tucker congruence ($r_{ex}*r_{em} > 0.99$ for all components in all cases). Wavelengths below 270 nm were excluded due to high leverage on models that led to noisy loading spectra and for ready comparison to the PARAFAC models presented elsewhere (Murphy et al., 2018). The full data set of EEMs from all degradation experiments was then projected onto the 4-component model derived from SRNOM PPL. This allowed standardization of the fluorescence signal loss we wished to

model. Fluorescence intensity at the maximum of each component (Fmax) was normalized to the second data point in each degradation experiment time series, as the first points (collected immediately before lamp exposure) were often outliers with aberrant residuals after modelling fluorescence losses (e.g. Eq. 2 and 3).

Previous studies (Murphy et al. 2018, Del Vecchio and Blough 2002) used a bi-exponential model to describe fluorescence

loss during photo-exposure as described in Eq. 2:

$$f_t = f_L e^{-k_L t} + f_{SL} e^{-k_{SL} t} \tag{2}$$

where $f_t$, total fluorescence normalized to the first EEM collected after the solar simulator lamp shutter opened at time t, is the sum of two fluorescence fractions ($f_L$ and $f_{SL}$) undergoing decay at different rates ($k_L$ and $k_{SL}$) (Murphy et al., 2018; Timko et al., 2015).


We modified Eq. 2 to replace time t with cumulative photon dose, assuming lamp photon output is constant throughout each experiment. If it can be properly measured, using cumulative photon exposure instead of time as the independent variable in models of fluorescence loss may allow better comparison of parameters between experiments, researchers, and experimental setups. The model is given in Eq. 3:

$$f_P = f_L e^{-k_L P} + f_{SL} e^{-k_{SL} P} \qquad\qquad (3)$$

where $f_P$ is total normalized fluorescence after cumulative photon exposure P (in moles of photons). Other variables are the same as in Eq. 2. Photon dose estimations from nitrite actinometry can be applied to DOM irradiated under the same conditions if those conditions allow for optically thin solutions during exposure. The 1 mm pathlength spiral exposure cell we used should ensure optical thinness even in highly absorbent DOM solutions.


Results from fitting Eq. 3 are reported as four separate parameters: $f_L$, $k_L$, $f_{SL}$, and $k_{SL}$. However, $f_L$ and $f_{SL}$ are not independent as they should always sum to 1. They are expressed separately in our results because we believe these f values may be useful for understanding the compositional bases of degradation differences despite the difficulties for interpretation this dependence presents, and because each f value was fitted separately, so modelled fits not always sum exactly to 1.


R software (v. 3.6.0) was used to fit bi-exponential models using the *nlsLM* function from the minpack.lm package, and R was also used for significance testing and plotting most results.

## 3 Results and discussion

We stated three goals of this study, claiming we would: 1.) identify methodological barriers to reproducible determination of
DOM photosensitivity and offer experimental guidelines to improve studies of DOM photodegradation kinetics; 2.) test our approach on samples from various environmental settings to see if our derived metrics of photosensitivity might respond to variability in DOM composition; and 3.) analyze photosensitivity differences between different DOM sources in detail to better understand the links between DOM composition, environmental setting, and photochemical degradation processes. Our results are presented and discussed in the same order. Section 3.1 discusses experiments using SRNOM that identify several sources
of experimental variability that influence photodegradation results which are crucial to apply our approach with confidence but also relevant to other methods of experimental DOM photodegradation. Section 3.2 shows that we were able to successfully apply our method to experiments using several different DOM sources. Finally, Section 3.3 presents a detailed comparison of experiments using samples from two freshwater wetlands to discuss the ecological relevance of photosensitivity differences measured with our approach. Sections 3.1 and 3.3 are further divided into topically distinct sub-sections for convenience.

**3.1 Method optimization and reproducibility**

**3.1.1 PARAFAC model**

Our results confirm many of the findings reported by Murphy et al. (2018) in that the fitted PARAFAC model of SRNOM PPL photodegradations produced similar components despite the independent data collection and analysis by different researchers (Fig. 1). Emission maxima for components 1 to 4 were 439, 412, 525, and 452 nm; however, only components 3
and 4 followed the bi-exponential decay pattern. Figure 2 shows an example of fluorescence change in each PARAFAC

component during photodegradation of SRNOM PPL. Component 3 in this study corresponds with $F_{520}$ in Murphy et al., 2018, while component 4 corresponds to the $F_{450}$. Matching component spectra to models in the online OpenFluor database confirmed these matches, with Tucker congruence r values over 0.98 for emission spectra for both components. The weaker match between component 4 in this study and $F_{450}$ in Murphy et al. is driven by differences in the excitation spectra (r = 0.949), but strong correlation between all 4 components in our PARAFAC model and higher information density in low wavelength ranges of excitation spectra could interfere with excitation spectral signal discrimination. Components 1 and 2 in this study did not exhibit bi-exponential decay during photodegradation. In most experiments component 1 decayed but did not follow a bi-exponential pattern, while component 2 showed little net change. Differences in PARAFAC component matches and behavior between this study and Murphy et al. (2018) could arise from operating at a different pH (3 here vs. their minimum pH of 4). For example, despite spectral differences, component 1 behaves similarly to $F_{420}$ in Murphy et al. (2018), which showed less rapid initial decay and a more linear overall pattern as pH decreased from 8 to 4 (see Fig. S4 in Murphy et al., 2018). Further results will focus on components 3 and 4 as they are most sensitive to photodegradation.

### 3.1.2 SRNOM experiments – experimental conditions and photon dose

Photodegradation kinetics in SRNOM trials were sensitive to many experimental conditions, but most importantly those that affected cumulative photon exposure. Key influences included volume of sample added to the irradiation system and DOM concentration, and we also tested for differences in results due to unknown discrepancies between individual researchers. Measurements made as a function of exposure time could obscure these differences if photon exposure was not instead directly estimated. In this sub-section we describe these methodological influences on results and demonstrate the utility of directly expressing results as a function of estimated photon exposure instead of exposure time.

Total volume of sample in the system affected degradation kinetics by altering the cumulative photon exposure relative to the abundance of optically active molecules. Figure 3 shows loss of absorbance at 254 nm and loss of fluorescence intensity of components 3 and 4 relative to starting values in experiments where total volume of sample varied. Sample volume predictably affects photon dose relative to the quantity of starting material, because in all trials a fixed volume of the total volume is exposed to light at any time before returning to the mixing vessel. We found that flow rates from 1.5 to 8 mL per minute did not impact photon dose (data not shown). Removing the magnetic stir bar in the equilibration vessel seemed to have a slight effect on absorbance and to a lesser degree fluorescence loss, so it was used throughout subsequent experiments. Expressing loss of absorbance and fluorescence as a function of estimated photon exposure rather than a function of time seems necessary to ensure comparability with other experimental systems, and we will follow this convention where possible.

However, the reader is reminded that actinometers do have limitations (e.g. broadband response measurement) and caveats exist for their successful interpretation. Because CDOM absorption spectra generally increase exponentially with decreasing wavelengths, many experimental designs may violate the requirement that samples are optically thin when irradiated (Hu et

al. 2002). The irradiation cell used here has a depth of 1 mm, which should prevent self-shading during photo-exposure at all concentrations tested. Previous work using this system showed that fluorescence loss was independent of SRNOM concentrations between 25 and 100 mg L$^{-1}$ (Timko et al. 2015). Concentration dependence in photochemistry is often assumed to stem from self-shading alone, and past work has shown the importance of working with "optically thin" solutions or properly correcting for inner filter effects when measuring photochemical behavior. All solutions shown here were considered optically thin at 300 nm and greater wavelengths following the convention that for optically thin solutions,

$$A_T \times L \ll 1 \tag{4}$$

where $A_T$ is total (Napierian) absorption coefficient and L is path length in m (Hu et al., 2002). Although inner-filter corrections can be applied to correct for self-shading in spectrophotometer cells with known geometry (Hu et al. 2002), these corrections cannot be easily applied in other irradiation designs (e.g. vials on their sides and spiral flow cells). The definition for optically thin solutions (Eq. 4) is somewhat vague, so we also tested the dependence of DOM concentration on photodegradation rates.

Degradation patterns seemed to be sensitive to DOM concentration as well but the effects were less clear (Fig. 4). In general, lower concentrations showed greater overall losses of absorbance and fluorescence. For the two most dilute solutions, PARAFAC C3 loss could not be modeled with a bi-exponential model, in contrast to all other samples throughout our study. Our results suggest either that our solutions experienced self-shading despite meeting the conventional definition of optical thinness, or some other mechanism links CDOM concentration to absorbance or fluorescence degradation kinetics such as concentration-dependent charge transfer interactions (Sharpless and Blough, 2014). Further work is needed to explain these findings.

Two researchers followed the same protocols with the same material (SRNOM PPL) as a test of reproducibility due to sample handling. Agreement between researchers was good and results varied to a similar degree as repeated tests by the same researcher (Fig. 5). Two-tailed t-tests were not able to distinguish differences in means between trials run by each researcher for any biexponential model parameters (p-values all greater than 0.10).

### 3.1.3 Effects of solid-phase extraction

Use of extracts vs. whole water samples is another major methodological choice that can affect results. Fluorescence degradation from reconstituted RO SRNOM and SRNOM PPL extracts generated the same PARAFAC components. However, the overall loss of modeled components 3 and 4 differed between SRNOM PPL extracts and RO SRNOM, as did kinetics of fluorescence loss (Fig. 6). The differences in fluorescence loss were small but systematic. Two-tailed t-tests of relative fluorescence loss suggested differences between PPL and RO SRNOM in PARAFAC component 4 (p-value < 0.01) with limited support for differences in component 3 (p-value = 0.06) and no support for differences in absorbance loss (p-value = 0.3 for 254 nm). Projecting the data onto a PARAFAC model built from RO SRNOM degradation data instead of SRNOM PPL data did not affect these results. Fitted model parameters from Eq. 3 suggest these differences stem from the kinetics of

the semi-labile fluorescence pool, with possible differences in the relative starting abundances of the labile vs. semi-labile pools (Fig. 7 and Table A1). Rate constants of the labile pool did not vary for either PARAFAC component, suggesting extraction did not affect behavior of this pool, so studies focusing on this pool should not be affected by PPL extraction.

Capturing changes in this pool is one of the explicit advantages of our experimental system, and future work on environmental photo-reactivity may focus on this time scale as photochemical reactions in the environment are often driven by initial rates (Powers and Miller, 2015). However, slower degradation processes or longer irradiations may be affected by extraction.

Shared PARAFAC components suggest PPL extraction did not strongly alter the compositional bases of fluorescence
photosensitivity in the RO SRNOM, but the differences in losses suggest researchers should take care when comparing extracts to original samples in future photodegradation kinetics studies. We are not sure what gave rise to these differences, but the RO SRNOM likely contains much more highly polar compounds such as (poly)saccharides and related compounds (e.g. glycosates). Differences between PPL and RO samples here are probably not due to variation in photon dose, as volume and initial absorbance were equal across samples. If concentration of fluorophores affects degradation kinetics, differing
fluorophore concentrations between our PPL extracts and whole SRNOM could explain the discrepancy. Even though we adjusted all samples to similar starting absorbance, selective enrichment or dilution of absorbing or fluorescing compounds in extracts could affect the mechanism responsible for any concentration dependence. Differences in electronic coupling and charge-transfer abilities (Del Vecchio and Blough, 2004; Sharpless and Blough, 2014) could arise in extracts and affect fluorescence degradation kinetics. RO SRNOM may present matrix effects relative to extracted SRNOM PPL, as metals and
other possible interferences are still present (albeit at much lower concentrations relative to DOC than in source water) despite the cation exchange and desalting treatments that accompanied the original reverse osmosis isolation (Kuhn et al., 2014).

### 3.1.4 Guidelines for photodegradation fluorescence kinetics experiments

It has been established that initial pH and pH change during photodegradation affects fluorescence photodegradation kinetics (Timko et al., 2015). We chose to conduct experiments at pH 3 because control by autotitration was not possible during these
experiments due to contamination from the pH probe, and starting at pH 3 ensured minimal pH change during photodegradation. If research goals do not explicitly include understanding effects of pH during photodegradation, we recommend bringing all samples to the same starting pH and controlling pH during the course of photodegradation experiments, or starting experiments at pH 3 and ensuring change during the experiment is minimal.

Using a reference material allows consistency within and between research labs. We recommend using SRNOM as it has been widely studied and characterized (Green et al., 2014). Comparing total absorbance and fluorescence loss and degradation kinetics of SRNOM to DOM sources of interest will allow more meaningful comparison between lab groups. Repeated experiments with the same standard can identify sources of error and quantify variability due to experimental procedures. Checking this variability against variability among repeated measurements of a sample may allow common variability to be

estimated and thus reduce the need for replication in future runs with similar DOM sources. We also used SRNOM (after solid phase extraction) as the basis for our PARAFAC model of fluorescence change during photodegradation and projected this model onto the rest of our data set, standardizing fluorescence losses between DOM sources to the same signal.

For research into compositional changes in DOM during photodegradation, test materials should be brought to similar starting
absorbance. We adjusted all samples to a raw absorbance of 0.12 at 300 nm (with a 1 cm path length), but this may be difficult or less ecologically meaningful with naturally dilute (e.g. ocean) or concentrated (e.g. leachates) DOM sources. If possible, testing different DOM concentrations for the same sample is recommended in order to establish any concentration dependence on photochemical rates. In our system,

Photon dose obviously affects degradation kinetics. Our experimental system offered several procedural choices that could affect photon dose, including volume of sample in the system and lamp intensity. Researchers should carefully control these parameters and ensure their procedures are generating reproducible results by running several replicated experiments with a reference material. We encourage repeating this process with multiple individuals within a lab to understand the impact of individual methodological choices on results (e.g. gravimetric measurement of volume added vs. pipetting, preparation of
samples). We strongly encourage at least reporting actinometry results or assumed actinometry for the experimental conditions used in order to better compare photon doses across studies and in the environment. While the additional work of actinometry is not trivial, we believe this represents one way to improve reproducibility of degradation kinetics that avoids the limitations of using time alone. Even this approach could be improved – our actinometer did not directly measure radiation across the UV spectrum, which could allow more accurate quantification of cumulative photon dose. Striking a balance between effort
required and reproducibility is difficult, but we believe our work illustrates some of the limitations of conventional approaches where photon exposure cannot be reliably calculated, and we hope our efforts inspire alternative approaches to overcoming these limitations. Ideally, samples should be irradiated under optically thin conditions when actinometry measurements or other approaches can be used to estimate photon doses for kinetic modelling (e.g. using Eq. 3 instead of 2).

Photodegradation is affected by both DOM composition and matrix conditions. While we found that the same PARAFAC model captured fluorescence decay in both SRNOM and solid phase extracts of SRNOM (as in (Murphy et al., 2018)), extraction did affect total fluorescence loss and its kinetics. However, we chose to use extracts for further experiments as in accordance with our research priorities and because our samples were not stable when stored as whole water samples. Preliminary experiments showed storage of water samples for greater than two weeks led to changes in fluorescence loss
patterns, even when filtered to 0.2 μm (see Appendix B). We believe this could have been due to the high DOC concentrations in samples used in those experiments, which could have been more susceptible to flocculation (von Wachenfeldt and Tranvik, 2008; Wachenfeldt et al., 2009) or other aggregation processes than dilute samples, but further work would be required to test this. While other work has found DOM absorbance remained stable in seawater samples after storage at 4° C up to 1 year

(Swan et al., 2009), concentrated DOM in inland waters may be unstable in cold storage conditions, affecting its optical properties or responses to photoirradiation. Further work is required to understand the cause of this behavior, but losses of DOC and changes to optical properties during cold storage of samples have been reported elsewhere (Peacock et al., 2015). We recommend using extracts with greater storage stability to allow comparison over time, unless all experiments can be conducted shortly after sample collection or previous experience shows that the optical properties of the DOM in question are stable for the duration of storage. As our goal was to test photosensitivity arising from DOM composition itself and not the effects of matrix chemistry, extraction was also conceptually appropriate. The tradeoffs and advantages of using whole water vs. extracts may be different in other experiments, and comparisons should probably be made to contextualize results when using extractions. Comparisons of kinetics between extracts and whole water samples should be made with care, but experiments using such comparisons may help disentangle the role of DOM chemical composition from other matrix effects in determining photodegradation behavior and sensitivity. Matrix effects may be especially important for extrapolating lab photodegradation findings to inferences at ecosystem scales. For example, if the approach described here is used to investigate longitudinal changes in DOM photosensitivity along a river network, tying these findings to residence times and photon doses in the field would be difficult without considering light attenuation by inorganic chromophores and particles. Matrix constituents may also fundamentally alter the photosensitivity of DOM by participating in charge-transfer processes. We recommend using DOM isolated from its matrix by extraction here not because it is a sufficient approach to understand these phenomena, but as a foundation to explore this complexity. More work is needed to understand the relative influence of DOM and matrix compositions on photodegradation kinetics.

## 3.2 Photosensitivity differences between DOM sources

After establishing procedures to understand and control experimental influences on DOM photosensitivity, our comparison of photodegradation of several DOM sources sought to reveal differences in photosensitivity arising from DOM. Figure 8 shows the degradation of PARAFAC components 3 and 4 relative to starting intensities in samples from different DOM sources. Both components showed potentially divergent decay patterns among DOM sources, with *Sargassum* leachate starkly diverging from bulk DOM sources. Fitted biexponential model parameters of decay in PARAFAC components 3 and 4 are shown in Tables A2 and A3, with parameters from component 4 plotted in Fig. 9 (similar plot for component 3 can be found in Appendix A, Fig. A2). We did not conduct repeated trials with every DOM source shown here due to logistical constraints, but t-tests on three trials each with SRNOM and one of the wetland samples supported potential differences in $f_L$ and $f_{SL}$ in both PARAFAC components, and possible differences in $k_{SL}$ in component 3. Notably, these two DOM sources had biexponential parameter values that were among the most similar compared to other sources (see "Small wetland" and "SRNOM" in Fig. 9), which suggests that our approach is sensitive enough to detect small differences.

The outlier in our comparison of DOM sources was *Sargassum* leachate extract, which was expected given the unique composition and the presence of phlorotannins (Powers et al., 2019). The natural DOM used in a previous study (Murphy et al., 2018) that yielded PARAFAC components appearing in all photodegradation experiments did not include leachates, only natural bulk DOM. Interestingly, this sample alone showed little or very slow semi-labile fluorescence loss with total fluorescence loss of projected PARAFAC components 3 and 4 dominated by rapid initial loss. Future studies using leaf or soil/sediment leachates, or lysed algal cells, or other putative sources of natural DOM instead of bulk natural DOM itself need to test this modelling approach more thoroughly to ensure it is appropriate, but using other leachate sources may highlight the compositional basis of the semi-labile fluorescence decay that seems ubiquitous in bulk natural DOM but absent in *Sargassum* leachate here. These experiments demonstrated the general applicability of our method to compositionally distinct DOM sources.

### 3.3 Photosensitivity and ecological inference

#### 3.3.1 Interpreting biexponential model parameters

Differences in biexponential model parameters between samples may allow reproducible comparisons of natural DOM photosensitivity. While the differences in parameter values described in Section 3.2 were encouraging, we wanted to know more about the potential ecological relevance of these differences. This approach has been used before given the excellent fit of this type of model to photodegradation data sets, and biexponential models indeed provided excellent fits to fluorescence losses in PARAFAC components 3 and 4 in our data sets. The biexponential model represents the sum of two terms, often referred to as labile and semi-labile to reflect the large relative differences in exponential slopes ($k_L$ and $k_{SL}$ in Eq. 2). This model captures loss of two pools of fluorescence intensity, possibly arising from two pools of DOM fluorophores decreasing in abundance at differing rates, or perhaps a single pool of photoreactive DOM with differing capacities for two types of reactions contributing to loss of fluorescence (Murphy et al., 2018).

In other studies (e.g. Murphy et al., 2018; Timko et al., 2015) the rate parameters $k_L$ and $k_{SL}$ have received the most attention, as different average rates of change in fluorescence governed by these rate constants may indicate differences in DOM chemical composition, matrix composition, environmental conditions (if experiments are performed in situ), or experimental conditions, making these values potentially useful metrics of compositional differences between DOM sources. However, differences in loss of fluorescence between samples may also arise from differing relative abundances of two "pools" of whatever is responsible for fluorescence at the beginning of the time series. Figure 10 shows degradation time series from two experiments, along with fitted model parameters. These experiments compare DOM sampled in October 2017 from the two freshwater wetlands in Maryland. Figure 10 shows loss of PARAFAC component 3 (see Fig. A3 in Appendix A for a similar plot showing loss of component 4). The model fits are shown against the data in upper panels, while the modelled fits for each of the two terms from Eq. 3 ($f_L e^{-k_L P}$ and $f_{SL} e^{-k_{SL} P}$) are plotted separately against the data in lower panels. This visualization

is useful to weigh the contribution of differing rate parameters ($k_L$ and $k_{SL}$) against the relative abundance of their respective fractions ($f_L$ and $f_{SL}$) at the onset of the experiment in determining overall differences in photodegradation behavior between samples. Component 3 loss models show similar $k_L$ values but different relative fractions of the "fast" pool of fluorescence loss at the start of the experiment. Differences in these starting fractions between samples may play a role in overall differences in degradation kinetics in component 4 as well. It is crucial to note that the chemical interpretation of these modelled fits is not clear. "Pools" of fluorescence in different relative abundances that decay at different rates may not map directly onto different groups of fluorophores changing in concentration. This behavior may stem from differences in the capacity for two classes of photochemical reactions – where k describes the reaction rates and f describes the relative capacity of the sample to undergo the corresponding reaction at the outset of the experiment. For example, one possible explanation offered for biexponential decay in Murphy et al. (2018) invoked reactive species degrading a single pool of fluorophores quickly and direct photolysis degrading those fluorophores more slowly. Differences in $f_L$ may then reflect differing capacity to form or react with triplet excited DOM. Further work is needed to understand what gives rise to relative differences between f terms in different samples, though as noted $f_L$ and $f_{SL}$ are not independent in the model presented here. This highlights one of the strengths of our approach – the ability to capture optical properties of DOM that change very quickly during photodegradation. The modelled labile portion of fluorescence contributes negligibly to total fluorescence after receiving between 0.5 and 1.2 moles of photons per square meter, (3-10 hours of irradiation with our experimental setup). Future work relating the photon dose required to reach this point and the environmental conditions affecting this dose in natural DOM could improve knowledge of DOM origins, residence times, and interactions with other degradation processes.

### 3.3.2 Linking photosensitivity to DOM sources in dynamic ecosystems

High resolution photodegradation experiments of natural DOM can reveal fundamental photophysical behavior of ecological importance. We believe the approach described here can help unravel sources or light exposure histories of DOM in natural settings. One of our overall goals is to determine relative photosensitivity among samples. The biexponential models that fit experimental photodegradation data may help with these comparisons. For example, in the two wetland samples compared in Fig. 10, distinct patterns of photodegradation suggest distinct DOM composition. DOM fluorescence in the larger wetland had relatively less "fast" decaying fluorescence in photosensitive PARAFAC components (parameter $f_L$) than the smaller wetland. These wetlands are depressions located less than 100 m from each other, but with isolated surface water during the October 2017 sampling. They differ in basin size, canopy cover, and vegetation communities. Our data and fitted model parameters suggest that DOM in the larger wetland has either previously been exposed to sunlight that has depleted the potential for "fast" decaying fluorescence, or that differences in source material or other processing of DOM pools in each wetland have given rise to relatively less photosensitive material in the larger wetland. In winter, water levels rose in each depression, and eventually both depressions were connected by surface flow from the larger to smaller wetland. Photosensitivity differences show DOM composition and reactivity are affected by these phenomena. Figure 11 compares biexponential model parameters in samples from each wetland depression taken in October 2017, December 2017, and April 2018 (component 3 shown here,

component 4 shown in Appendix A, Fig. A4). This is an especially dynamic period in the seasonal cycles that affect DOM in this area – the October sampling is just before deciduous leaves senesce and fall, and the December sampling occurred less than a month before rising surface water levels connected the two depressions. Figure 11 shows that we may be able to capture the effects of ecosystem phenomena on DOM sensitivity. $k_L$ values for both PARAFAC components do not show any obvious pattern, while $k_{SL}$ values are very similar at each sampling site for all three dates but may be changing between dates due to some shift in DOM composition over time affecting both sites. The most obvious pattern is in $f_L$ and $f_{SL}$. These differ between sites in October and December, suggesting that despite their proximity, conditions at these sites differ enough to affect DOM photosensitivity in their surface water. The larger depression shows less faster-decaying fluorescence, either due to differences in the source of the material on the landscape or depletion relative to the smaller depression reflecting greater light exposure and natural degradation. These differences are homogenized in April, when surface water mixing (and shorter residence times in surface storage due to export) means site-specific processes are less influential in shaping DOM composition.

These photosensitivity differences may have consequences for other ecosystem processes. For example, if low $f_L$ at the time of sampling reflects high rates of photodegradation in wetland surface water, photopriming may contribute to microbial heterotrophy. Or wetland DOM with high $f_L$ may influence downstream ecosystems, if DOM exported to stream networks is then susceptible to photodegradation which alters its lability to heterotrophs (Judd et al., 2007) or promotes flocculation (Helms et al., 2013). The sensitivity of our approach may also allow revisiting questions of longitudinal dynamics of light exposure in stream systems (Larson et al., 2007).

We can use this example to justify the effort involved in modeling fluorescence decay kinetics by comparing these inferences to those possible with simpler approaches. Modeling kinetics of fluorescence loss allows us to resolve processes apparently occurring at different rates, obvious in the large differences between $k_L$ and $k_{SL}$ in any of the samples analyzed. Differences in $f_L$ values between samples provide novel information and are the basis for our comparison of photosensitivity to relatively fast photochemical processes. These parameters can only be derived from time series of measurements collected frequently. Otherwise our basis of comparison is differences in overall changes to fluorescence and absorbance. In the wetland samples described here these differences were small or nonexistent and showed no discernible patterns. An example is shown in Fig. 12, which shows absorbance spectra before and after irradiation in samples from the two wetlands (using only two of the dates shown in Fig. 11 for visual simplicity). Samples show similar behavior in all cases, or differences due to lack of resolution producing errors (as in large apparent relative changes in the small wetland Nov. 2017 sample, which arise from small changes near the limit of detection). Additionally such comparisons are difficult to compare between studies – any change is contingent on photon dose, which would either need to be replicated with identical experimental apparatus or related to only two time points, making calculations of rate prone to error. It might be possible to glean more information from the absorbance time series during irradiation, but this would require selecting optimal wavelengths to isolate for modeling losses or modeling losses at many wavelengths, approaches which to our knowledge have not been developed. Instead we can model kinetics of a

540 tractable number of variables (PARAFAC loadings) for each sample that provide novel information building on existing approaches to characterize DOM composition. Our approach is not the only way to compare photosensitivity and effects of photodegradation, nor will it be appropriate for all studies. We encourage all researchers to ensure their approaches address the methodological issues raised in Section 3.1, but the advantages of the specific data and modeling employed by our method correspond to our research questions and may not be universal.

Photodegradation of DOM extracts in the lab does not replicate *in situ* photodegradation of DOM in surface waters. However, in situ photodegradation of DOM in surface water is extremely convoluted – the complexity of DOM chemical composition, surface water matrix composition, simultaneous ecological processes that also alter DOM composition, and the natural dynamism of surface water systems are intertwined and make it difficult to understand the role of photodegradation of DOM

in surface water ecosystems. Our approach represents one step in the direction of disentangling this story but leaves many questions unanswered. We demonstrated several sources of potential variability in degradation kinetics that require more attention, any of which may affect our understanding of different influences on in situ photodegradation and its ecological consequences. Further research is required to understand how differences in DOM composition alone (as isolated in our work with extracts) interact with matrix composition (Grebel et al., 2009; Poulin et al., 2014; Timko et al., 2015; T. Stirchak et al.,

2019), and how these reactivity differences affect other DOM transformation processes (Amado et al., 2015; Chen and Jaffé, 2016; Lønborg et al., 2016) and ecosystem- or macrosystem-scale or biogeochemistry (Anderson et al., 2019; Pickard et al., 2017; Rutledge et al., 2010).

This example is not conclusive for these sites – though we demonstrate differences in photosensitivity between samples that

have plausible hypothetical causes and consequences for ecosystem processes, these remain speculative. Our example does illustrate the possible uses of our method. Clearly much more research is needed to explain the observed differences in photodegradation kinetics between these two wetlands and test these hypotheses, ideally with more detailed data on DOM composition associated with differing photosensitivity. Regardless, our approach can complement established techniques for describing DOM such as bulk optical properties, ultrahigh resolution mass spectrometry, or nuclear magnetic resonance, and

could be combined with other experimental approaches probing natural DOM sources and transformations.

**4 Conclusion**

Our research identified several methodological issues that can improve photodegradation experiments and leveraged this knowledge to show how photosensitivity differences may relate to DOM composition and environmental setting using a case study. Photodegradation experiments have improved our understanding of the role of DOM light sensitivity in ecological

processes. As researchers continue to explore related questions and experiments proliferate, it is important to use approaches that constrain the influence of experimental conditions and promise reproducible or at least comparable results. Our method

allows reproducible and relatively short experiments that capture photosensitivity differences between varying sources of natural DOM on time scales relevant for investigating degradation processes in the environment. This approach can be used to ensure experiments conducted at different times or by different researchers can be compared. Our work illustrates several obstacles to reproducing and comparing studies of photodegradation kinetics, highlights underappreciated sources of uncertainty, and offers an approach that improves upon past methodological limitations. It also captures distinct fast dynamics that differ between samples that would be lost in experiments measuring only total changes in optical properties or using far fewer time points. We explored the possibility of using this approach for inferences about ecosystem processes by comparing photosensitivity metrics between samples from two adjacent wetland areas, showing that photosensitivity differed in space and time in patterns that generated plausible hypotheses. Closer parsing of the biexponential decay parameters from modelled fluorescence loss may also allow differentiating DOM sources, past exposure to photodegradation, or future photodegradation potential in other ecosystem settings.

## Appendix A: Tables and additional figures

Appendix A includes figures and tables that complement information in the main text.

## Appendix B: Storage time experiment

We ran a series of experiments testing effects of storage time on photodegradation kinetics which are relevant to our overall results but were performed under different conditions. These preceded the other experiments and the experimental setup was modified based on their results. These used filtered water samples (not extracts) taken from the small and large wetland sites described above, but were collected in November 2017. They were filtered, stored in the dark at 4°C, and run through the photodegradation system undiluted using a 3x3mm quartz flow cell in the spectrofluorometer instead of the 10x10mm cell used for all other experiments. Experiments were run after 5-8, 9-13, and 14-16 days of storage. These results are reported as a function of time rather than photon exposure as no actinometry was collected with an analogous experimental setup.

## Data and code availability

Data and code used in this analysis are available from the Dryad repository at https://doi.org/10.5061/dryad.hmgqnk9d9 (Armstrong, 2020).

## Author contribution

AA developed the method's applications for ecological inference, collected the data, analyzed the data, and drafted and edited the manuscript. LP assisted in the method's conception, collected the data, assisted with data analysis, and edited the manuscript. MG conceived the method, designed and optimized the instrument system, assisted with data analysis, and edited the manuscript.

## Competing interests

The authors declare they have no conflict of interest.

## Acknowledgments

A. Armstrong was supported in part by US National Science Foundation Grant # DBI-1052875 and cooperative agreement # 58-1245-3-278 between the United States Department of Agriculture, University of Maryland, and the National Socio-Environmental Synthesis Center. Dr. Margaret Palmer provided valuable feedback during planning and manuscript preparation. Katherine Martin provided the sample from Parker's Creek included in the comparison of DOM source material. Jessalyn Davis assisted with actinometry measurements. This is contribution 6002 (CBL 2021-072) of the University of Maryland Center for Environmental Science, Chesapeake Biological Laboratory.

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

## Tables

**Table A1. Fitted biexponential model parameters (Eq. 3) for comparison between RO and PPL SRNOM. p-values are from two-sided t-test of difference in means; n = 3 for both RO SRNOM and SRNOM PPL. $f$ is unitless and $k$ is m$^2$ [mol photons]$^{-1}$.**

| PARAFAC component | Biexponential parameter | RO SRNOM Mean (SD) | SRNOM PPL Mean (SD) | t-test p-value |
|---|---|---|---|---|
| 3 | $k_L$ | 3.35 (0.252) | 2.63 (0.654) | 0.18 |
| 3 | $f_L$ | 0.0623 (0.00207) | 0.0411 (0.0102) | 0.065 |
| 3 | $k_{SL}$ | 0.133 (0.0150) | 0.177 (0.00792) | 0.02 |
| 3 | $f_{SL}$ | 0.936 (0.00113) | 0.960 (0.0107) | 0.058 |
| 4 | $k_L$ | 2.72 (0.110) | 2.39 (0.276) | 0.16 |
| 4 | $f_L$ | 0.155 (0.00717) | 0.140 (0.0139) | 0.2 |
| 4 | $kS_L$ | 0.132 (0.143) | 0.191 (0.00346) | 0.015 |
| 4 | $fS_L$ | 0.838 (0.00897) | 0.856 (0.0142) | 0.16 |

**Table A2. Fitted biexponential model parameters (Eq. 3) for different DOM sources. $f$ is unitless and $k$ is m$^2$ [mol photons]$^{-1}$. Where n > 1, means are shown with standard deviations in parentheses. SRNOM PPL trials are three experiments using the same sample source, small wetland trials include three trials with the same sample source (to test non-SRNOM system stability) and two trials with samples from other dates, and large wetland trials include one trial each from three sampling dates. p-values of biexponential model fits all below 1x10$^{-6}$ except for $k_{SL}$ for Sargassum, p = 0.016. Model parameters for every individual trial can be found in associated data set (Armstrong, 2020).**

| PARAFAC component | Biexponential parameter | SRNOM PPL (n = 3) | Stream (n = 1) | Coastal ocean (n = 1) | *Sargassum* (n = 1) |
|---|---|---|---|---|---|
| 3 | $k_L$ | 2.63 (0.654) | 3.17 | 2.84 | 5.4 |
| 3 | $f_L$ | 0.0411 (0.0102) | 0.106 | 0.135 | 0.11 |
| 3 | $k_{SL}$ | 0.177 (0.00792) | 0.237 | 0.137 | 0.0194 |
| 3 | $f_{SL}$ | 0.960 (0.0107) | 0.887 | 0.857 | 0.889 |
| 4 | $k_L$ | 2.39 (0.276) | 3.58 | 4.55 | 3.68 |
| 4 | $fL$ | 0.140 (0.0139) | 0.222 | 0.229 | 0.334 |
| 4 | $k_{SL}$ | 0.191 (0.00346) | 0.308 | 0.332 | -0.00803 |
| 4 | $fS_L$ | 0.856 (0.0142) | 0.764 | 0.762 | 0.653 |

**Table A3. Fitted biexponential model parameters (Eq. 3) for wetland DOM samples. $f$ is unitless and $k$ is m$^2$ [mol photons]$^{-1}$. Where n > 1, means are shown with standard deviations in parentheses. p-values of biexponential model fits all below 1x10$^{-6}$. Model parameters for every individual trial can be found in associated data set (Armstrong, 2020).**

| PARAFAC component | Biexponential parameter | Small wetland, 2017-10-05 (n = 3) | Small wetland, 2017-12-20 (n = 1) | Small wetland, 2018-04-01 (n = 1) | Large wetland, 2017-10-05 (n = 1) | Large wetland, 2017-12-20 (n = 1) | Large wetland, 2018-04-01 (n = 1) |
|---|---|---|---|---|---|---|---|
| 3 | $k_L$ | 2.86 (0.199) | 2.37 | 4.12 | 2.69 | 6.01 | 1.34 |
| 3 | $f_L$ | 0.120 (0.00908) | 0.126 | 0.0732 | 0.0281 | 0.214 | 0.0496 |

| | | | | | | | |
|---|---|---|---|---|---|---|---|
| 3 | $k_{SL}$ | 0.184 (0.0203) | 0.138 | 0.124 | 0.185 | 0.137 | 0.127 |
| 3 | $f_{SL}$ | 0.881 (0.00906) | 0.875 | 0.932 | 0.975 | 0.980 | 0.960 |
| 4 | $k_L$ | 2.65 (0.0255) | 2.58 | 3.20 | 2.03 | 3.60 | 2.73 |
| 4 | fL | 0.195 (0.00698) | 0.181 | 0.165 | 0.109 | 0.0711 | 0.152 |
| 4 | $k_{SL}$ | 0.176 (0.0126) | 0.165 | 0.110 | 0.190 | 0.175 | 0.132 |
| 4 | $fS_L$ | 0.798 (0.00820) | 0.814 | 0.833 | 0.887 | 0.928 | 0.844 |

## Figure captions

**Figure 1.** (a) Spectral loadings and (b) contour plots of PARAFAC components (1-4, left to right) modeled from EEMs of SRNOM PPL extract photodegradation time series. In the top row, dashed lines represent excitation spectra and solid lines show emission spectra in the top. The full dataset of all degradation time series EEMs was projected onto this model.

**Figure 2.** Example of fluorescence change in PARAFAC components during photodegradation. Data show degradation of SRNOM PPL.

**Figure 3.** Photodegradation time series of absorbance at 254 nm and fluorescence intensities of PARAFAC component 3 and 4 relative to starting values. Data are shown from experiments with SRNOM PPL that varied volume of sample added to mixing reactor (after filling flow cell lines). Top panels (a-c) show values as a function of exposure time, while bottom panels (d-f) show values as a function of cumulative photon exposure calculated from $NO_2/NO_3$ actinometry.

**Figure 4.** Photodegradation time series of (a) absorbance at 254 nm and (b, c) fluorescence intensities of PARAFAC component 3 and 4 relative to starting values. Data are shown from experiments with SRNOM PPL that varied approximate DOC concentrations. In all experiments 0.5 ml SRNOM PPL solution was added to mixing reactor after filling flow lines.

**Figure 5.** Photodegradation time series of PARAFAC component 3 (panel a) and 4 (panel b) fluorescence intensity, relative to starting values. Data are shown from experiments using SRNOM PPL performed by 2 of the authors to test reproducibility of results.

**Figure 6.** Photodegradation time series of PARAFAC component 3 (panel a) and 4 (panel b) fluorescence intensity, relative to starting values. Data are shown from 3 replicates of both RO SRNOM and SRNOM PPL.

**Figure 7.** Fitted biexponential model parameters (Eq. 3) from the time series of loss of PARAFAC components 3 and 4 in irradiation experiments comparing RO SRNOM to PPL SRNOM (see Fig. 6 for data). f is unitless and k is $m^2$ [mol photons]$^{-1}$. C3 and C4 denote PARAFAC components 3 and 4. Error bars represent mean $\pm$ standard deviation from three experiments. Two-tailed t-tests suggest differences in $k_{SL}$ for both components ($p = 0.020$ in component 3, $p = 0.015$ in component 4), while $f_L$ and $f_{SL}$ may differ ($p = 0.065$ and 0.058) in component 3. (a) $k_L$, (b) $ks_L$, (c) $f_L$, and (d) $fs_L$.

**Figure 8.** Photodegradation time series of PARAFAC component 3 (panel a) and 4 (panel b) fluorescence intensity, relative to starting values. Data are shown from experiments using PPL extracts from different DOM sources (see Methods for source descriptions). "Large wetland" and "Small wetland" samples use the same symbol for samples from each source, including samples collected on different dates.

**Figure 9.** Fitted biexponential model parameters (Eq. 3) from the time series of PARAFAC component 4 (see Fig. 9 for data). f is unitless and k is $m^2$ [mol photons]$^{-1}$. For wetland samples, shapes represent different sampling dates (circles are 2017-10-04, triangles are 2017-12-20, and squares are 2018-04-01).

**Figure 10.** Data and model fit of PARAFAC component 3 loss in experiments with two wetland samples collected October 2017. Top panels show data and model fit (Eq. 3) while bottom panels decompose the fitted model into its two summed terms, $f_L e^{-k_L P}$ and $f_{SL} e^{-k_{SL} P}$, or labile and semi-labile terms. (a) Data and fit for small wetland, (b) data and fit for large wetland, (c) data and decomposed model for small wetland, and (d) data and decomposed model for large wetland.

**Figure 11.** Fitted biexponential model parameters (Eq. 3) from the time series of PARAFAC component 3, comparing DOM from large and small wetland sampling sites collected on different dates. f is unitless and k is $m^2$ [mol photons]$^{-1}$.

**Figure 12.** Changes to absorbance spectra after photodegradation show advantages of high-resolution fluorescence data. (a) Initial vs. final spectra (calculated as absorption coefficients) in photodegradation experiments using samples taken from the two wetlands on two of the dates also shown in Fig. 11. (b) Same data, but expressed as percent change in absorbance across the spectrum.

**Figure A1.** Photograph of photoirradiation system with key components labeled. A micro gear pump circulates sample between the equilibration chamber, a flow cell cuvette inside the spectrofluorometer, and the spiral exposure flow cell underneath the solar simulator lamp. A water bath set to 25°C (outside picture) circulates water through the larger black tubing to the glass jacket surrounding the pictured water bath in which the equilibration chamber sits.

**Figure A2.** Fitted biexponential model parameters (Eq. 3) from the time series of PARAFAC component 3 (see Fig. 9 for data). f is unitless and k is $m^2$ [mol photons]$^{-1}$. For wetland samples, shapes represent different sampling dates (circles are 2017-10-04, triangles are 2017-12-20, and squares are 2018-04-01).

**Figure A3.** Data and model fit of PARAFAC component 4 loss in experiments with two wetland samples. Top panels show data and model fit (Eq. 3) while bottom panels decompose the fitted model into its two summed terms, $f_L e^{-k_L P}$ and $f_{SL} e^{-k_{SL} P}$, or labile and semi-labile terms. (a) Data and fit for small wetland, (b) data and fit for large wetland, (c) data and decomposed model for small wetland, and (d) data and decomposed model for large wetland.

   **Figure A4.** Fitted biexponential model parameters (Eq. 3) from the time series of PARAFAC component 4, comparing DOM from
large and small wetland sampling sites collected on different dates. f is unitless and k is $m^2$ [mol photons]$^{-1}$.

   **Figure A5.** EEMs of filtered source water samples compared to reconstituted solid phase extracts. (a) Large wetland, source water. (b) Small wetland, source water. (c) Large wetland, soild phase extract. (d) Small wetland, solid phase extract. Figure shows samples originally collected 2017-10-05.

   **Figure B1.** Time series of photodegradation experiments on whole water wetland samples. (a) Change in PARAFAC component 3
over time, relative to initial intensity; (b) Change in PARAFAC component 4 over time, relative to initial intensity. Each column represents samples from a different wetland source. Three experiments were run with aliquots drawn from a water sample from each wetland, and results seemed to change with storage time. First experiments with each wetland water source were run 5-8 days after sample collection, second experiments were run 9-13 days after sample collection, and third experiments were run 14-16 days after collection. The trend in most cases toward lower relative photosensitivity in measured variables and in some cases increasing
data noise (or poor PARAFAC model fit) as samples aged informed the decision to use solid phase extracts to improve reproducibility.

**Figures**

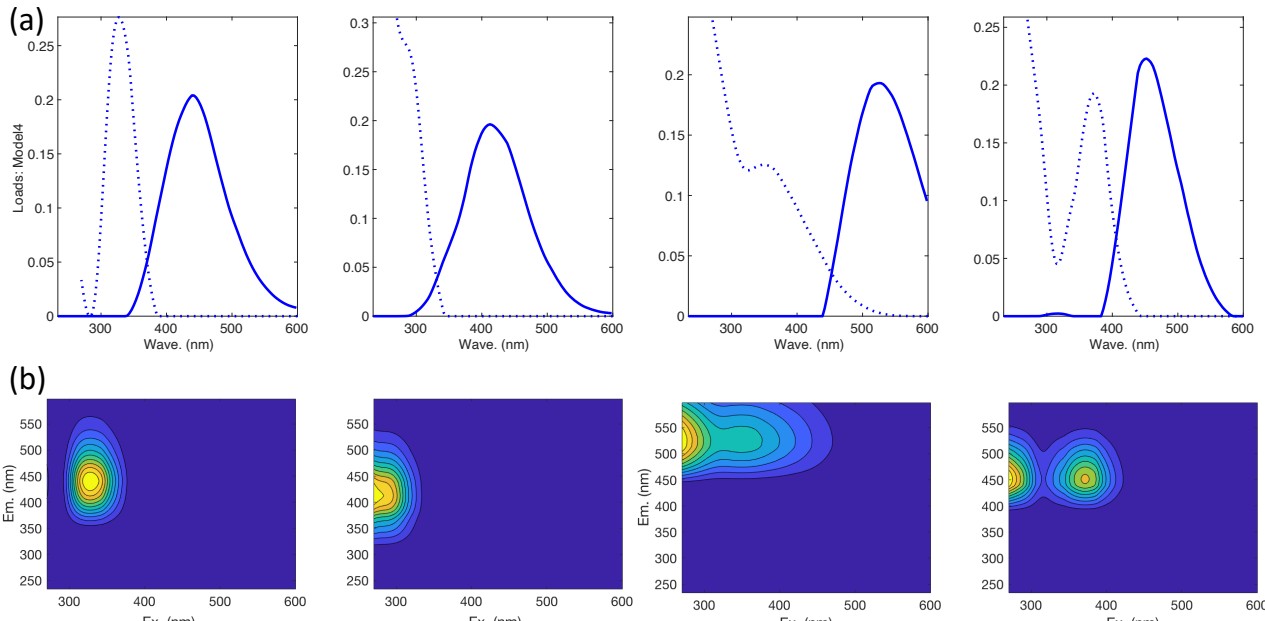

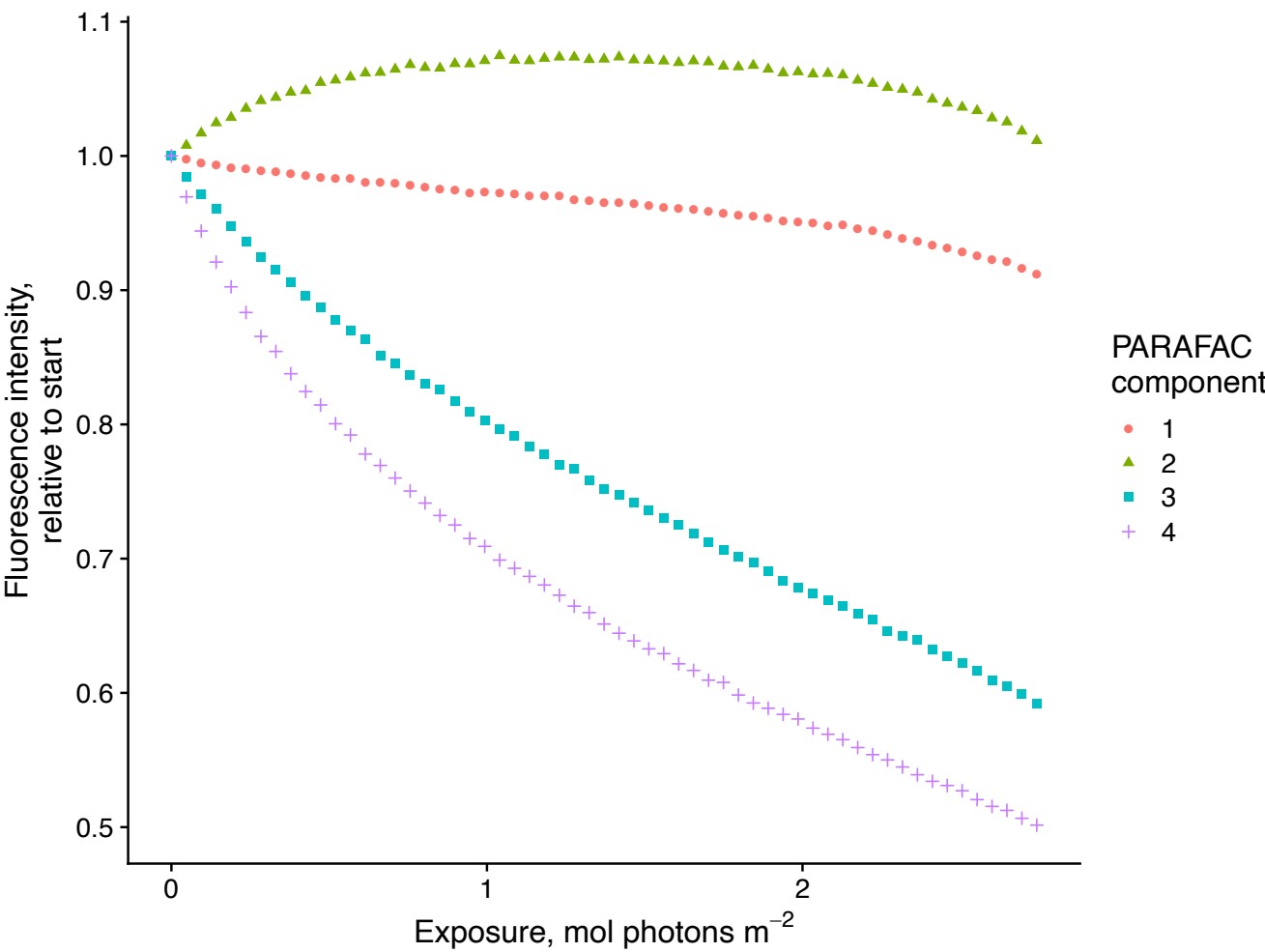


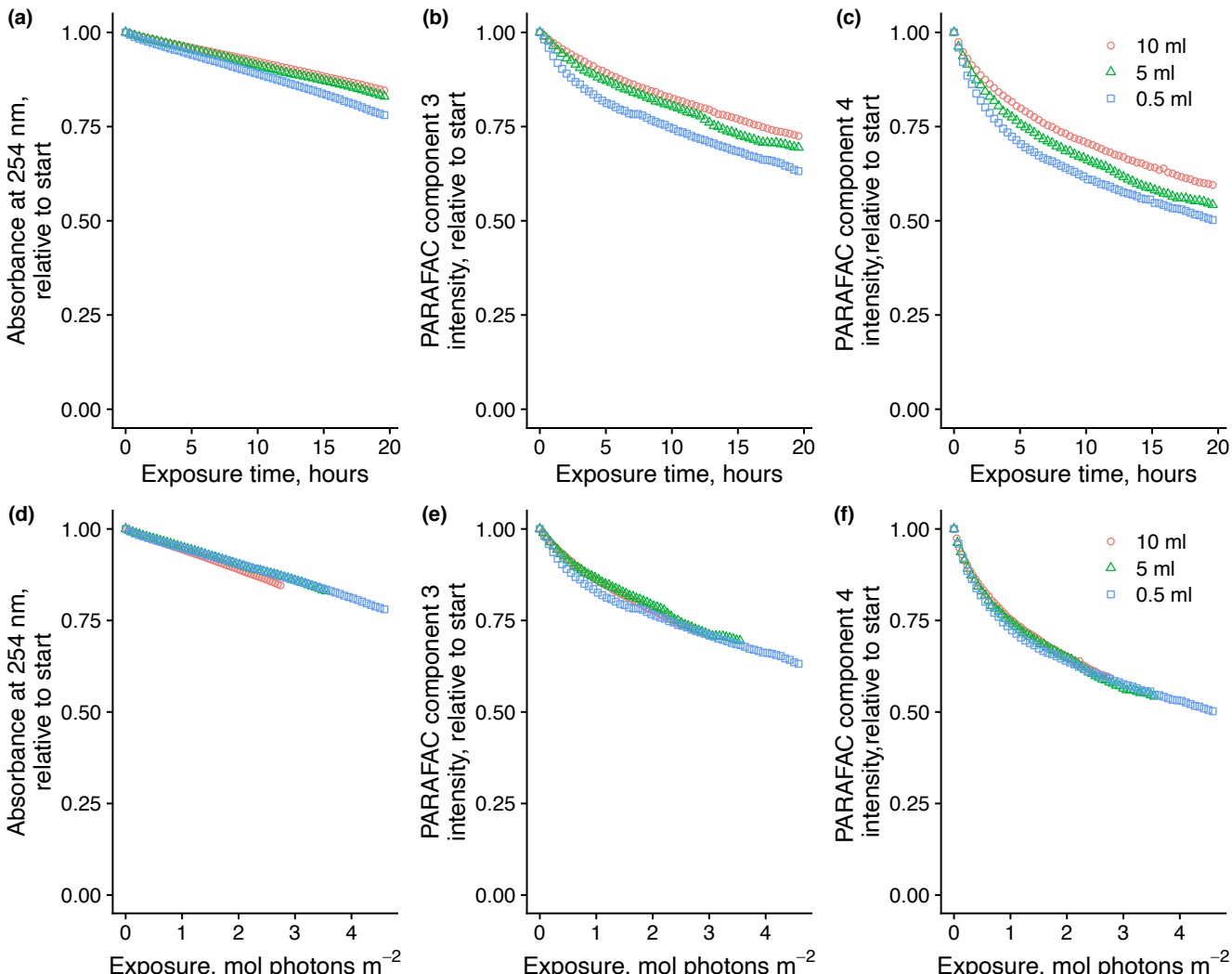

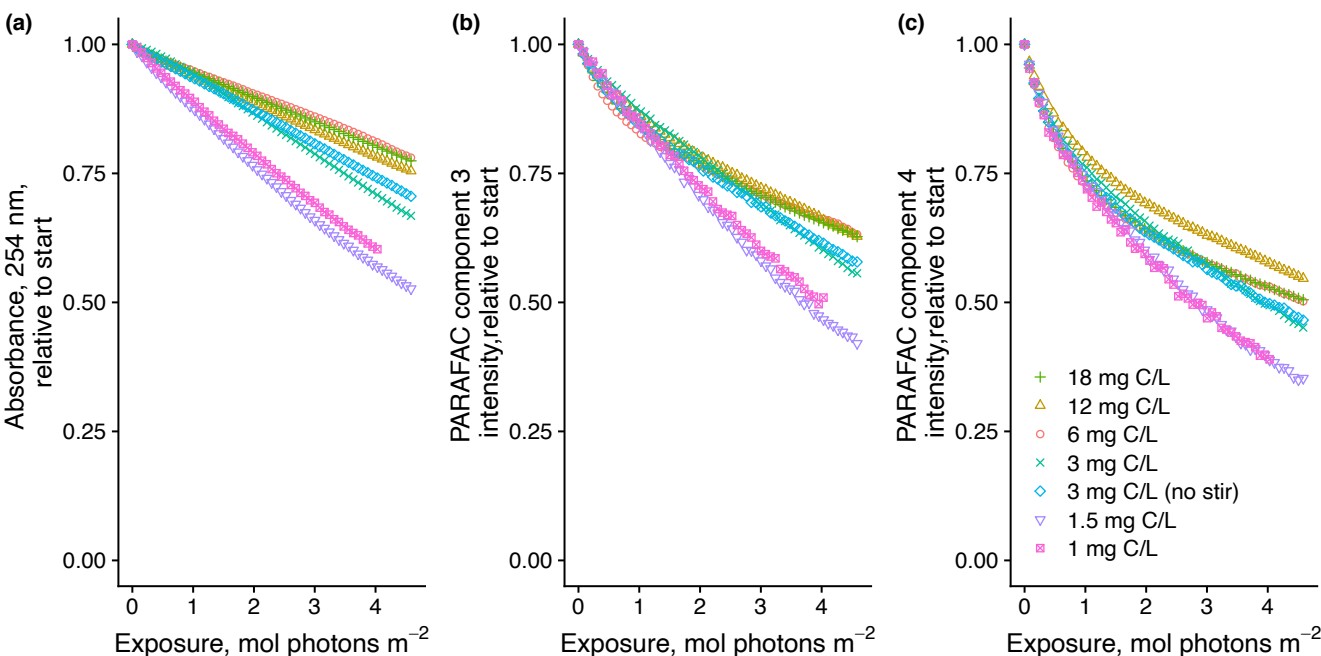

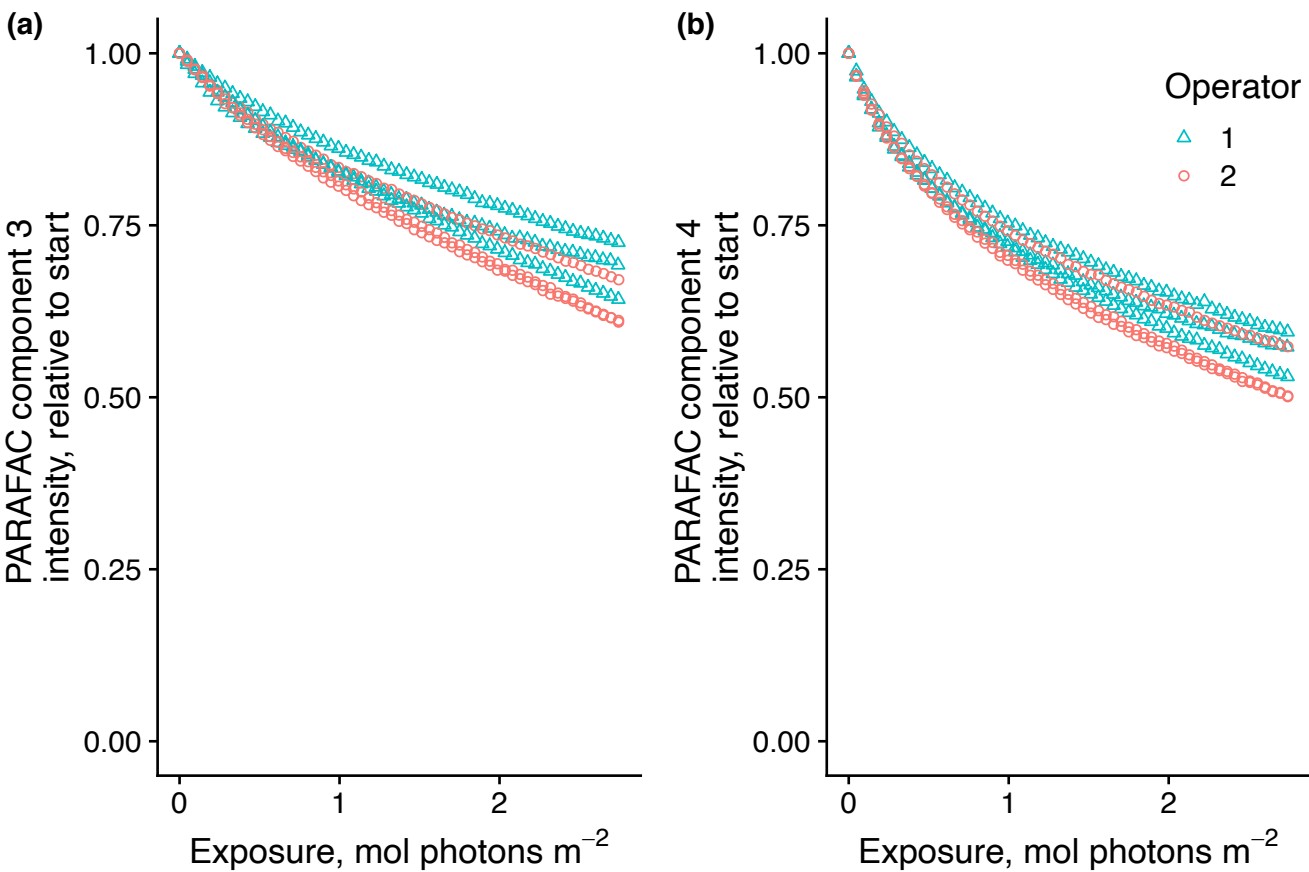

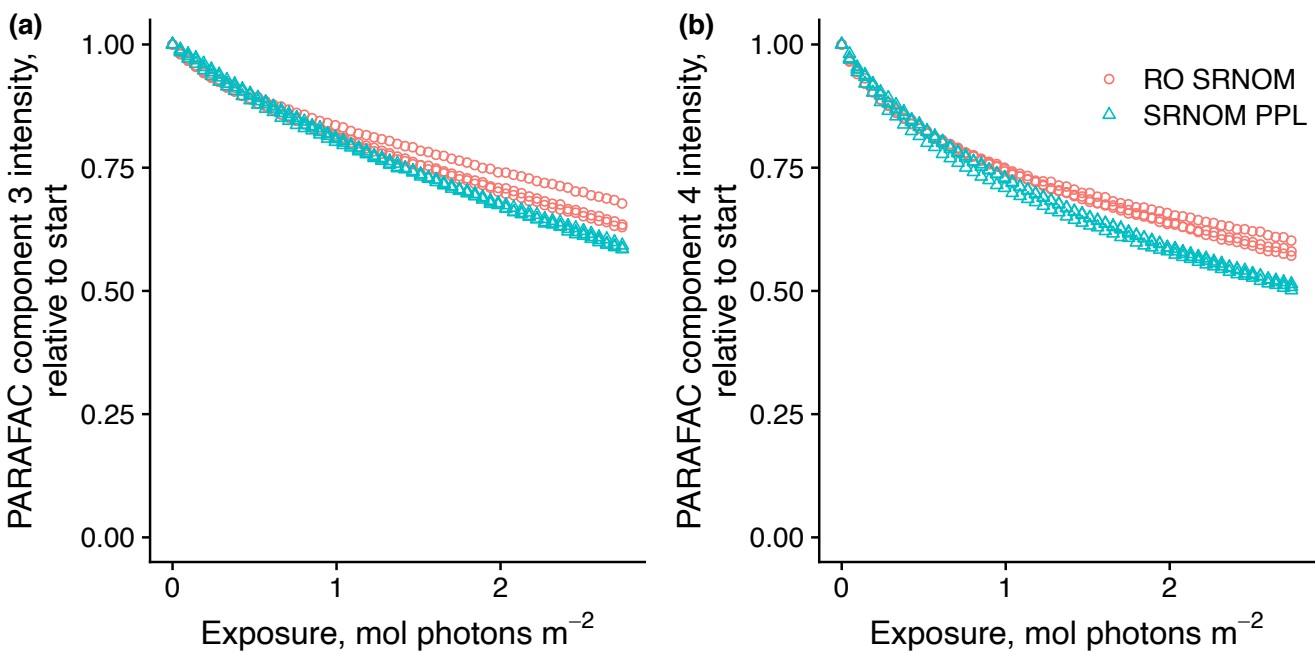

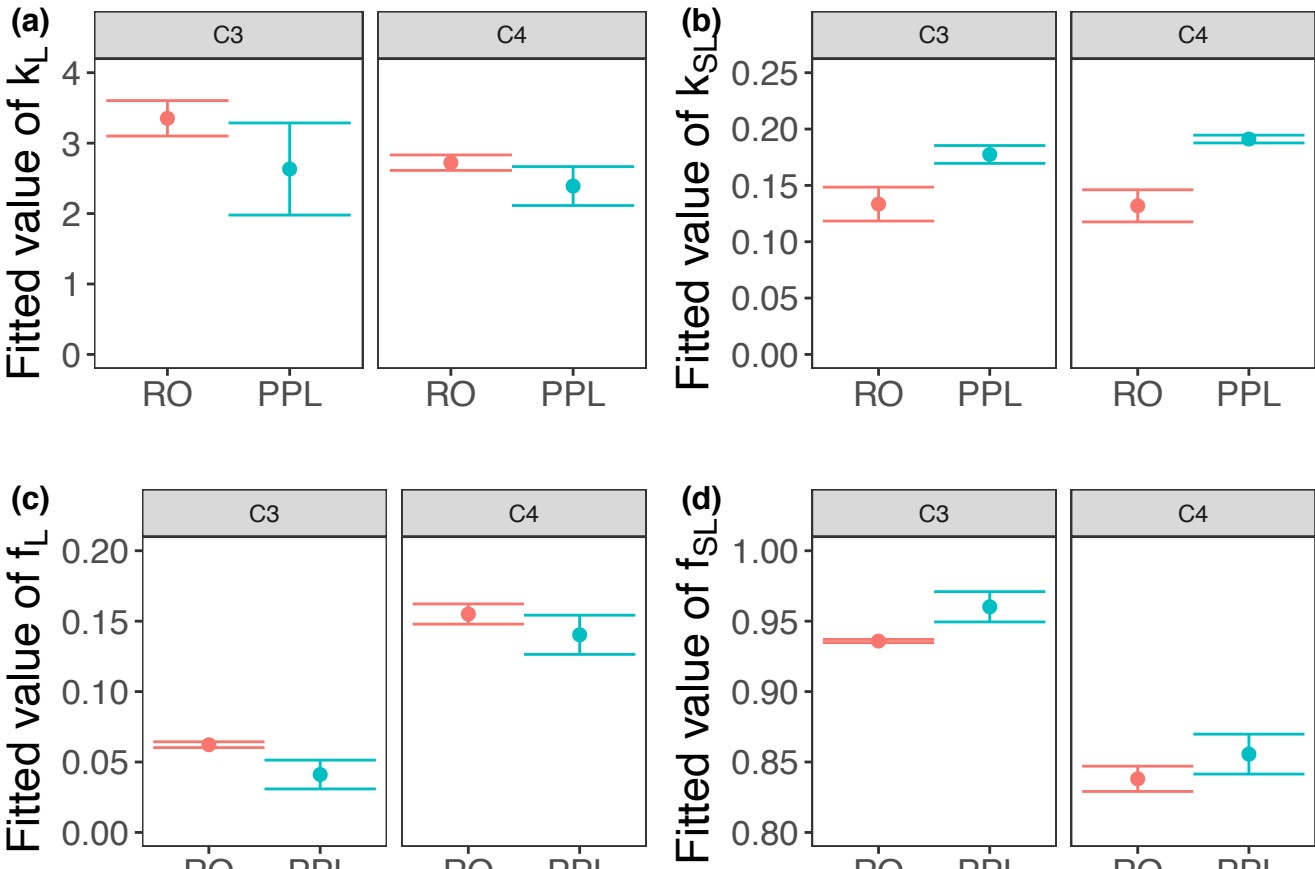


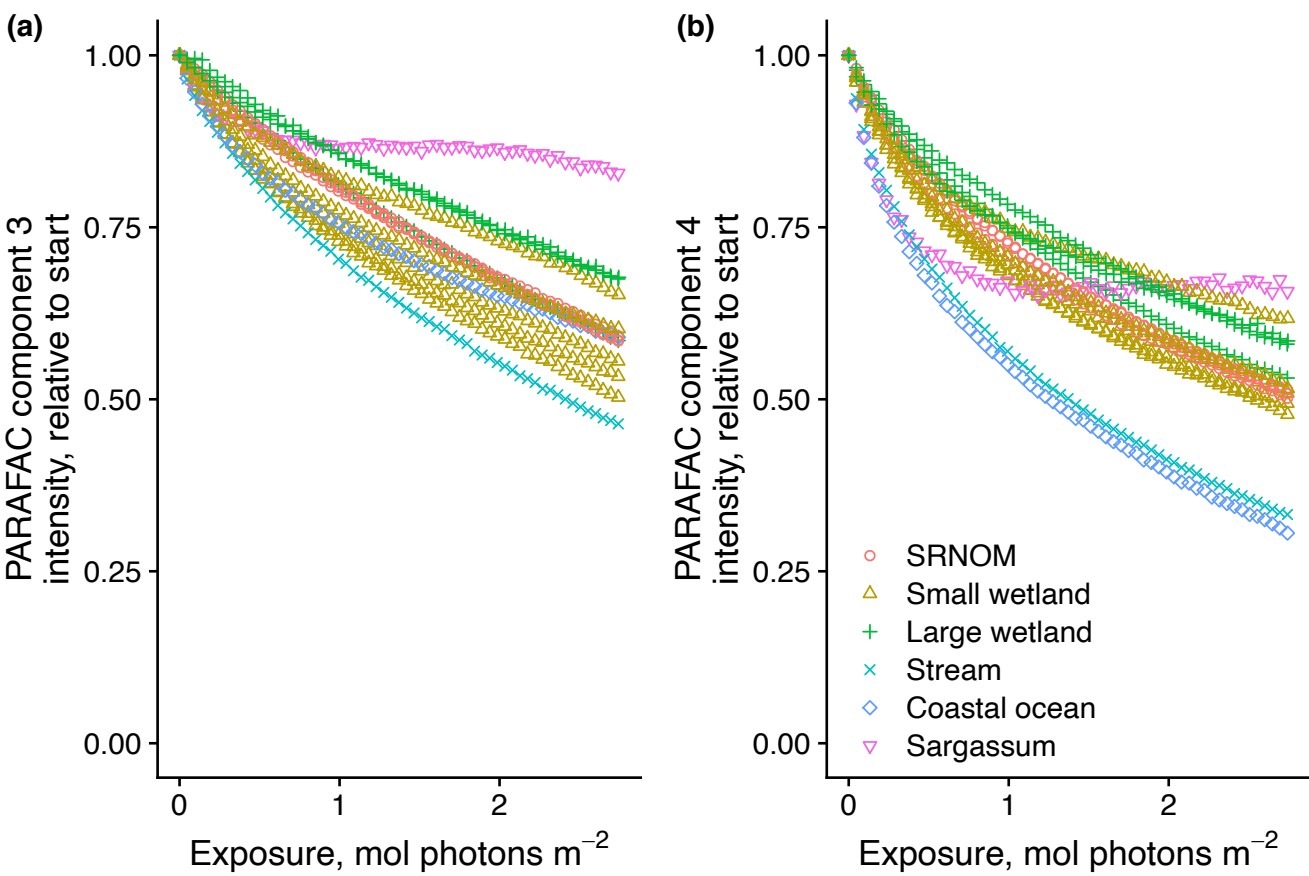

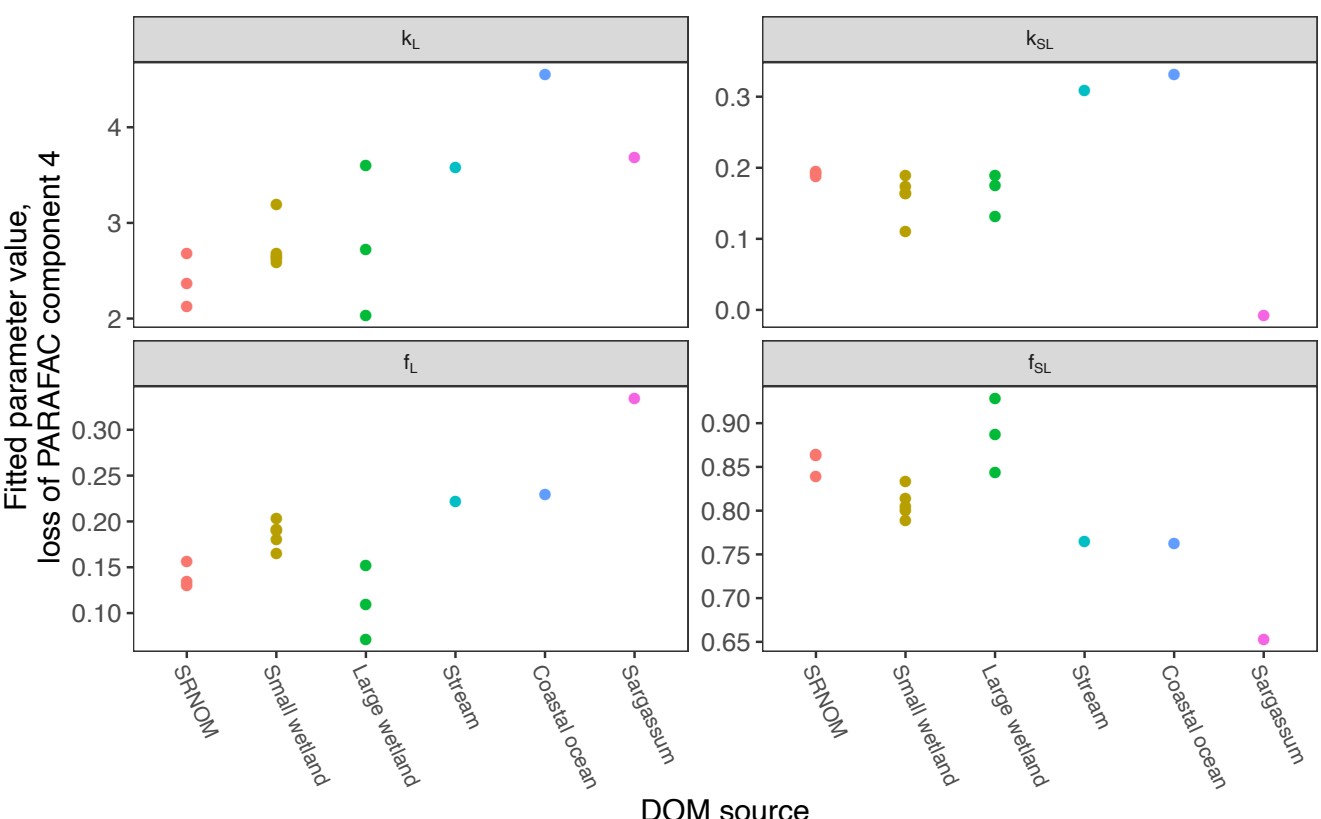

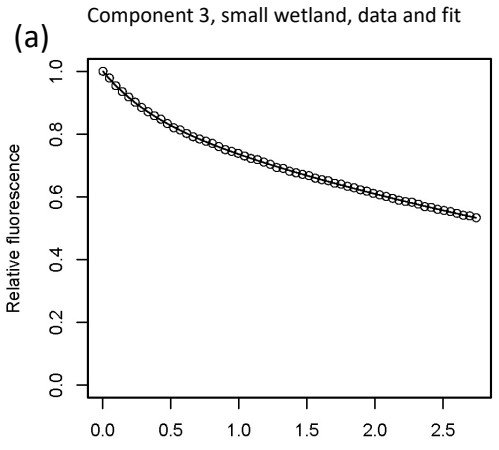

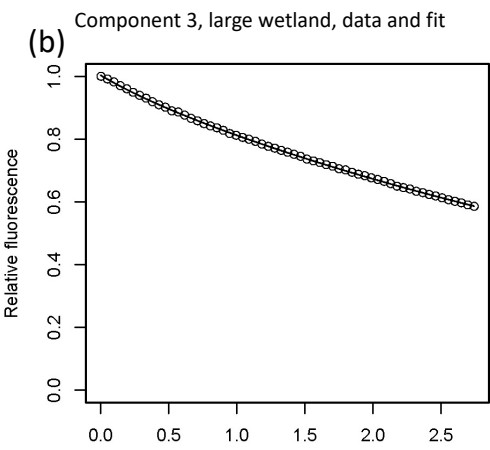

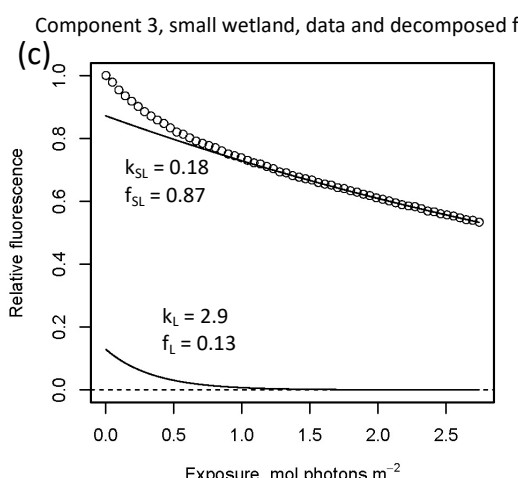

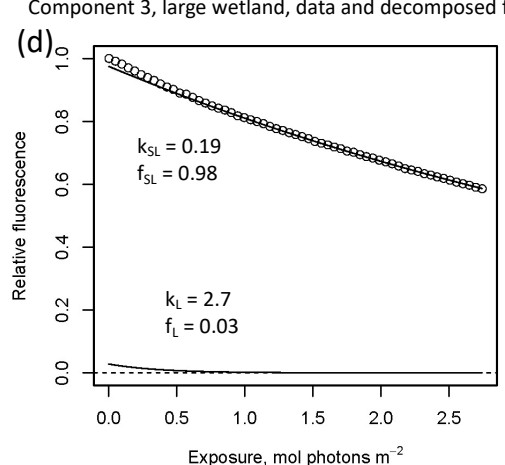

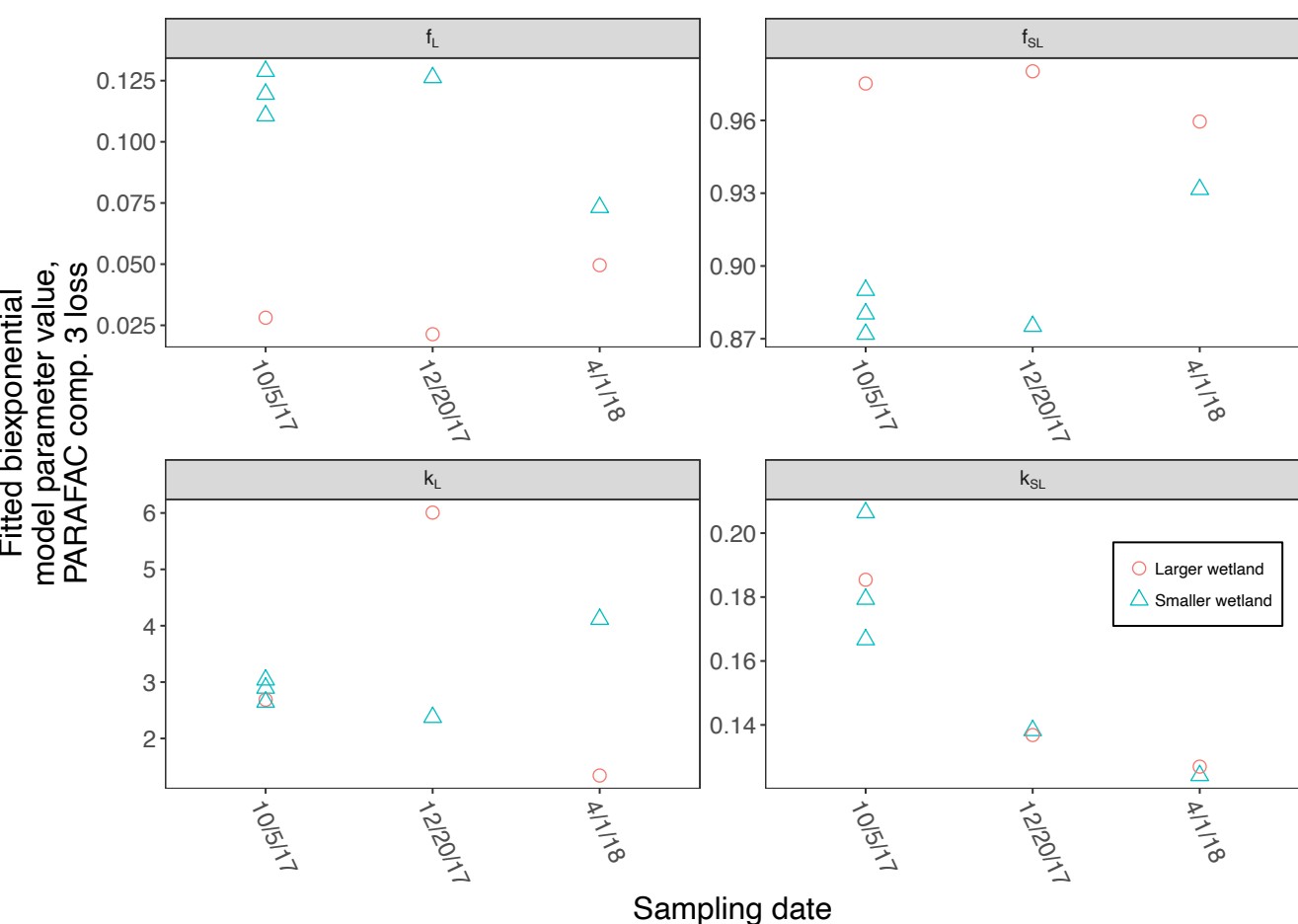

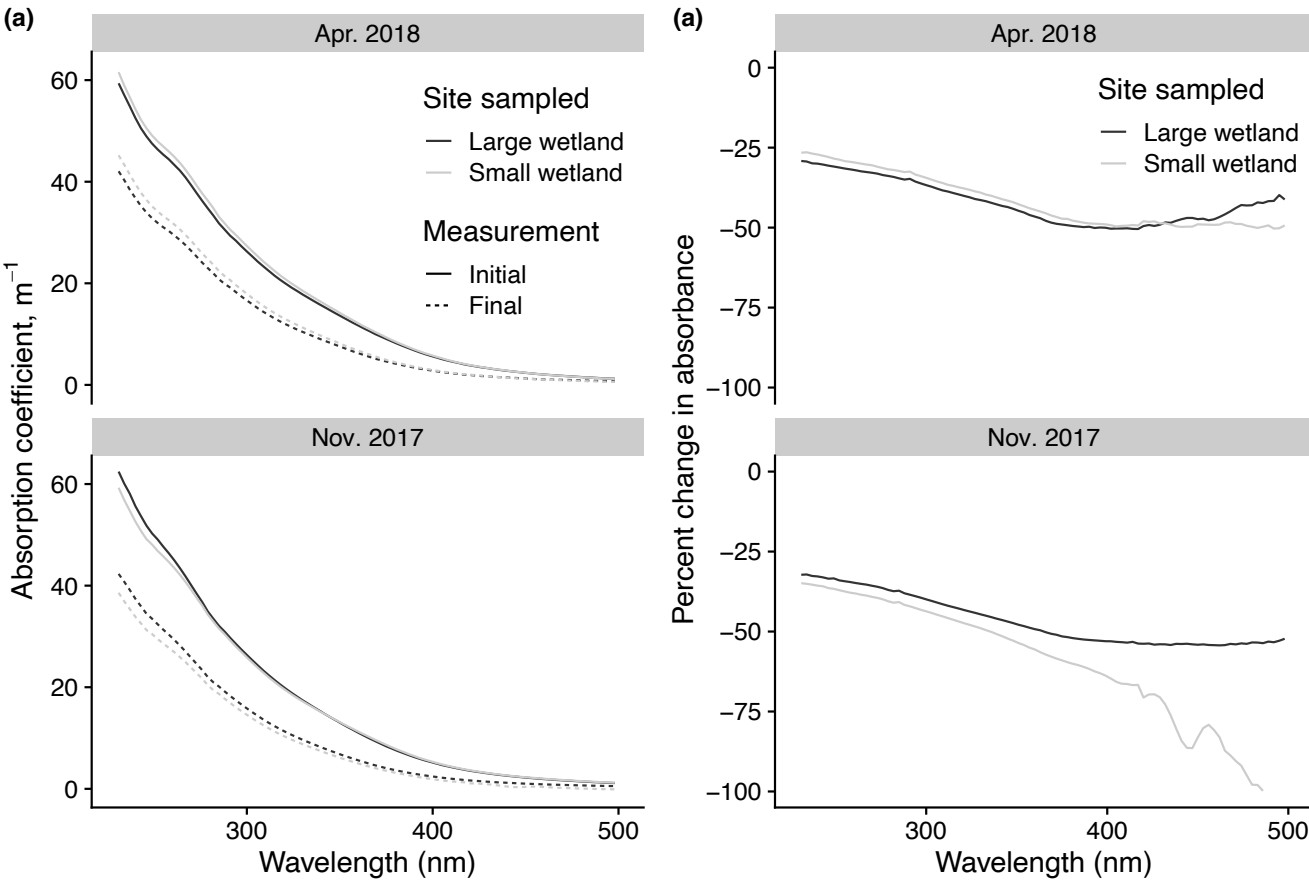


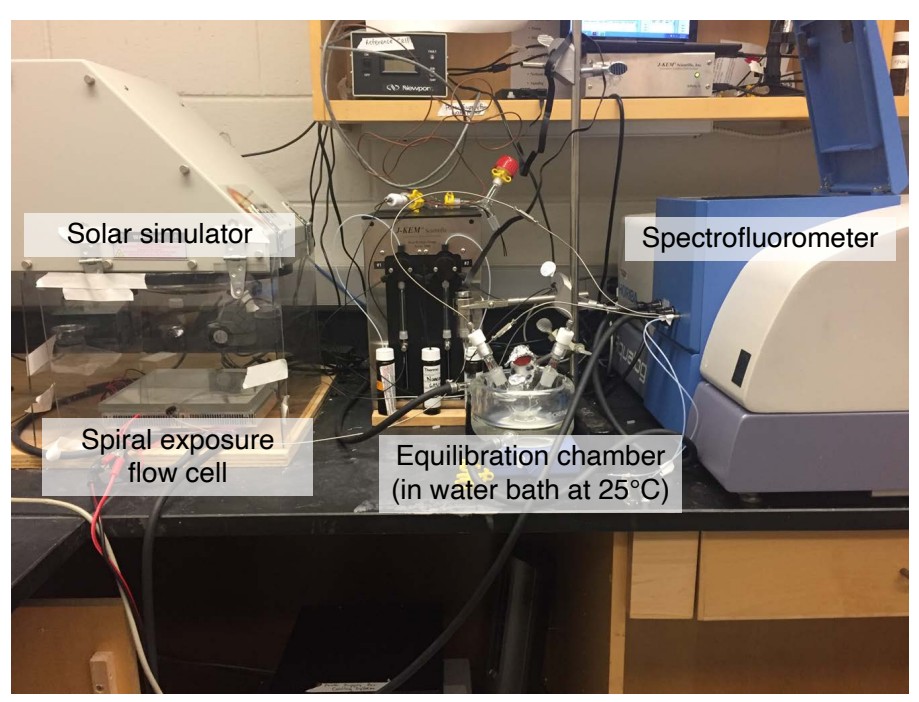

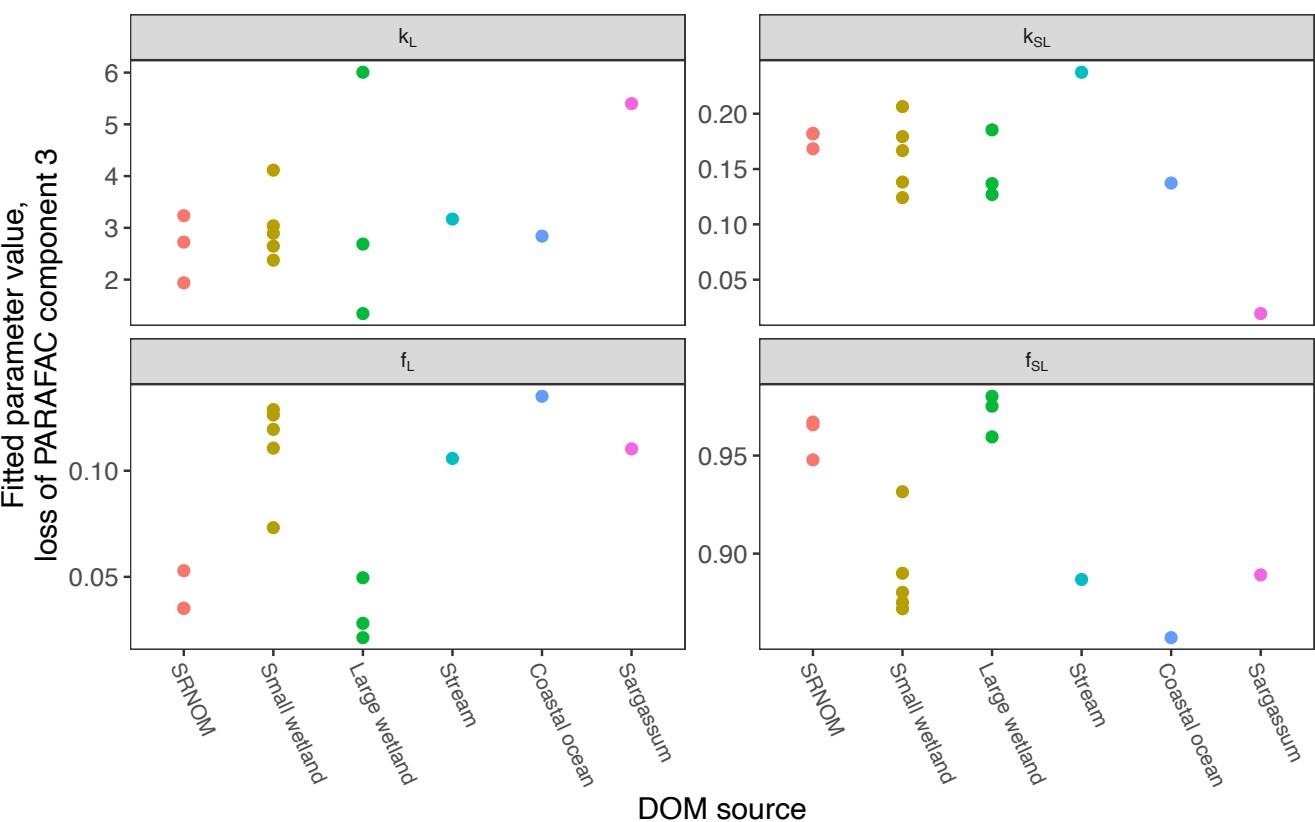

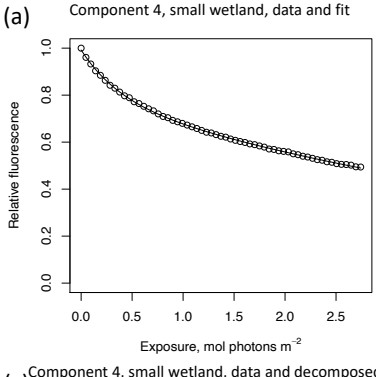

(a)

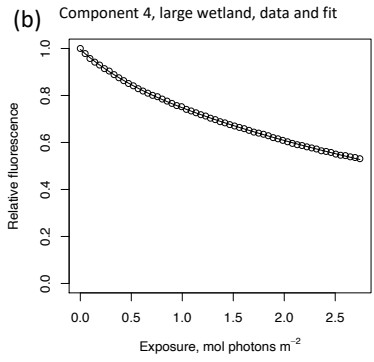

(b)

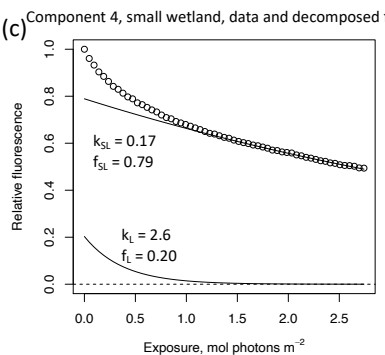

(c)

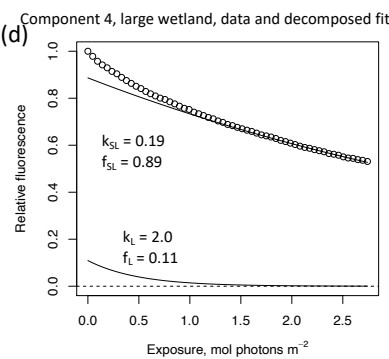

(d)

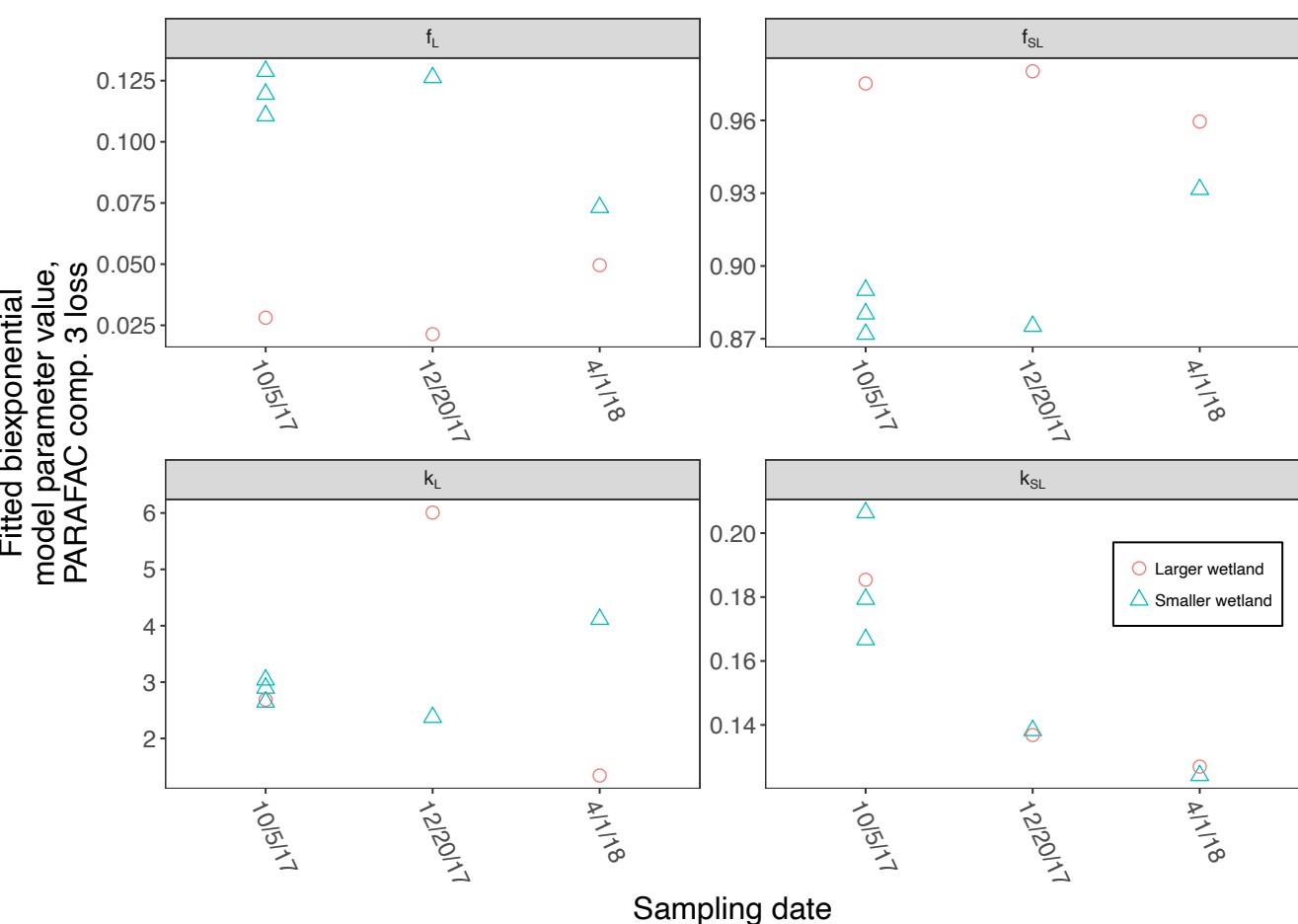

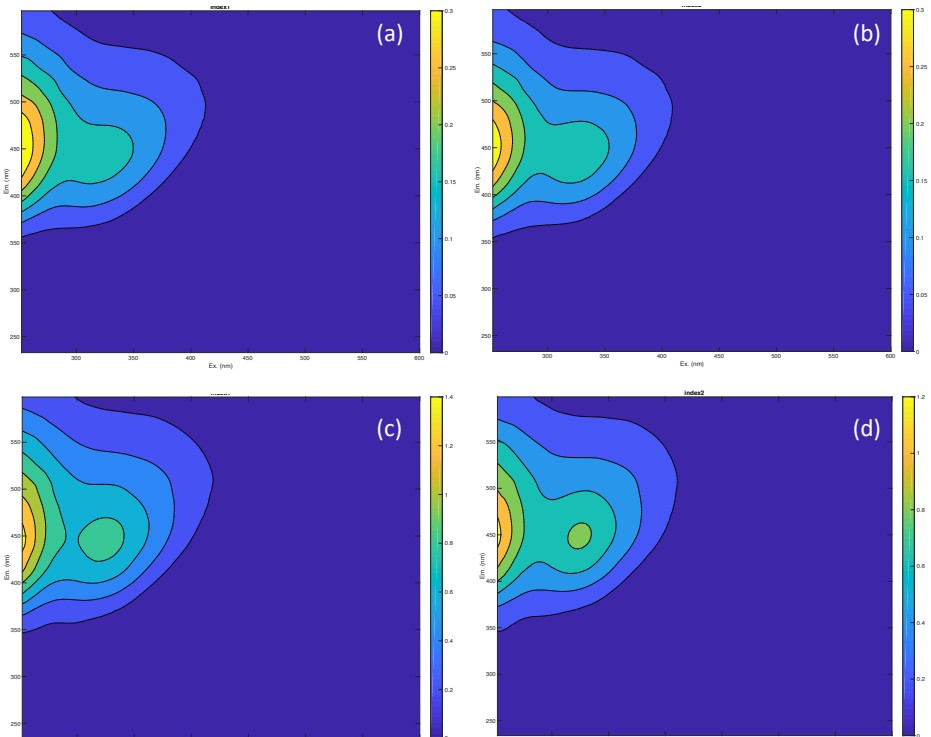


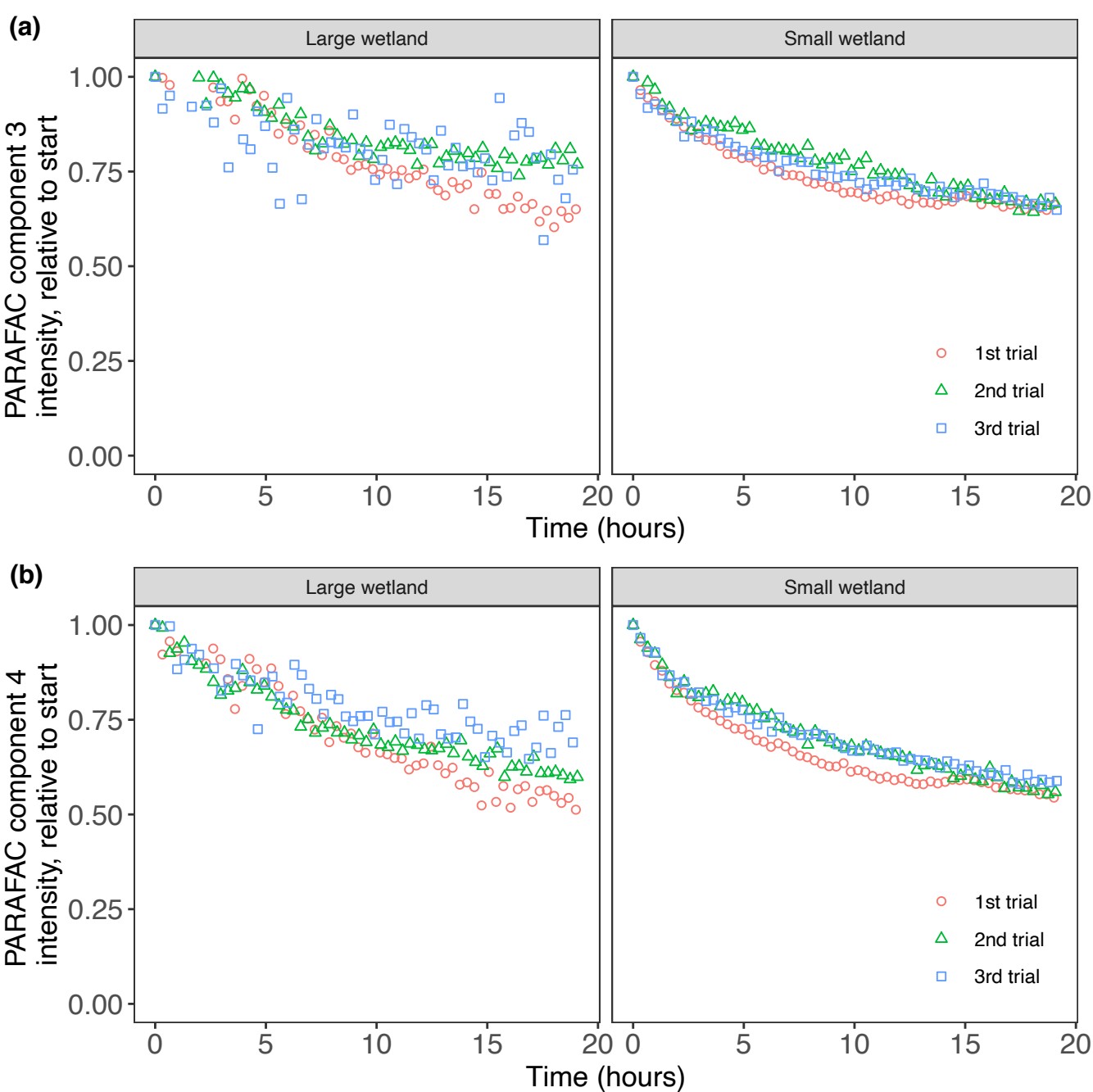