# Peer review of "Reproducible determination of dissolved organic matter photosensitivity"

_Biogeosciences, 2020_

## Referee Comment (RC1) · Patrick Neale (Referee) · 30 Jul 2020

General Comments

Measurements of spectral fluoresence have become a standard part of any study of the dynamics and distribution of dissolved organic matter in both freshwater and marine environments. Methods for deriving standardized metrics from fluorescence data are well established, and most studies of DOM dynamics acquire detailed sets of excitation-emission matrices and analyze them using parallel factor analysis (PARAFAC). In the present contribution, Armstrong, et al., argue that the kinetics of fluorescence photosensitivity are also useful properties to characterize DOM, in particular, the magnitude and rate constants of biexponential kinetics. However, several important considerations

need to be observed if these photosensitivity metrics are to be compared across studies. These are shown in the context of a specialized continuous loop exposure/assay system that the authors have devised, but several are applicable to any photosensitivity study. One is the use of a reference material, SRNOM, to both test the derivation of PARAFAC components and standardize the kinetics for a given exposure set up. Another informative test is the reproducibility of the results in repeated runs including those by different observers, which helps to establish the uncertainty in the estimated kinetic parameters.

I quite agree with these recommendations which are straightforward to implement. For others, it is not clear that they will always contribute to reproducible, or more importantly, interpretable, results. One is the preparation of solid phase extracts of DOM vs using fresh filtrates for the measurements. While extracts are inherently more stable than raw filtrates, the authors demonstrate differences in the photosensitivity kinetics of extracts vs the original filtrates, the reasons for which are not well understood. The authors advise preparing extracts in some situations but not others, this leaves open the question of comparability. The authors also advise using a standard optical density in the sample, yet also show there is a concentration dependence to the kinetics, even for an optically thin exposure conditions. Again, a decision will have to made of whats more important, knowing how the DOM behaves at its natural concentration or being able to compare it to other sources.

The authors also advocate expressing kinetic results as a function of cumulative photon exposure as opposed to time, this will adjust for differences in exposure regime between different studies. In principle, this is a good idea. Any photosensitivity study should include a detailed documentation of the optical setup and exposure measurements such that the results can be expressed either as a function of time or cumulative exposure. Studies do range in how thoroughly exposure conditions are documented. For these kinetic studies, exposure has been measured with a nitrite actinometer. Actinometry is very useful to quantitate the effects within a specific optical set up and is

used effectively by the authors to monitor exposure between experiments, including those with variable working volume. However, the nitrite actinometer only measures quantum fluxes between 330 and 380 nm. For incident solar UV, this is only about 60 per cent of the spectrum (by quanta). For comparison across all photosensitivity studies, I would suggest a more general exposure metric, e.g. total UV radiation. Given the broadband 330-380 nm quantum flux, total UV can be estimated with reference to the manufacturer's stated spectral distribution. But lamp and component aging can change that (even adjusting for constant power), so the best approach is to measure the spectrum in conjunction with actinometry (admittedly, neither of these are trivial to perform).

In the end, investigators will need more information to decide whether the additional insight gained from defining photosensitivity kinetics will be worth the efforts needed to perform all these standardization steps. This report documents experimental tests that show what are sources of variability in the procedure but does not show what new understanding is actually being gained from the results (vs what can already be gained from other optical/chemical measurements typically made on DOM). As far as reproducibility, there is only one instance of repeat determination on a "real world" sample, the small wetland. Further repeat determinations are needed to demonstrate the general reproducibility of the approach. Also, some modification of how the photosensitive fractions are estimated may be needed if the results are to be used in a comparative context (see specific comments below). Despite that this is still a work in progress, it is quite appropriate to have the authors' study and recommendations in a Biogeosciences discussion article. Comments by a wide range of readers should help refine the ideas proposed and ultimately advance the field.

Specific comments:

Line 140 "Solar exposure" – This was not solar exposure, but a laboratory set up. Better (?) "Total sample exposure"

Figure 7 and Table 1 – The captions for these describe kL and kSL with units of mol photons m$^{-2}$. Those are the units of P, the units of k should be the inverse, m$^2$ [mol photons]$^{-1}$.

Figure 7 and Table 1 apparently show the same data – and both state the statistics of the comparison. Questionable whether both are necessary. Figure 7 caption, however, says see Fig. 5 for data. It appears that Fig. 6 has the data.

Line 350-360 – Factors affecting storage of filtered water. Previous text only mentioned preparation of filtrates using 0.7 $\mu$M glass fiber filters, which allows passage of a substantial bacterial fraction. However, subsequent guidelines caution against used stored material even filtered to 0.2 $\mu$M (some bacteria can even get through those). Please clarify what filtration method was used for the stored sample, but in any case (slow) microbial degradation may be another factor resulting in changes in stored material.

Figure 10 – Caption says see Figure 6 for data, data appears to be in Fig. 9. This correction also applies to the caption for Figure A1. Figure 10 could be misleading to the reader, as a quick inspection may suggest that the spread of the multiple points for a particular sample source (and standard deviation in the table) are indications of reproducibility. Actually, only the SRNOM and three of the small wetland points are repeat determinations on the same sample. Separating the points for different sample dates would avoid this confusion, especially given that later on (section 3.3.2) the authors attribute some of the difference in the wetland results with time to possible differences in sample composition and/or previous solar exposure.

Line 456ff, section 3.3.1 Here the relative abundances of kinetic components are considered as metrics of DOM composition or other influences on photosensitivity. A problem with the comparative use of fL and fSL is that the two parameters are not independent. Assuming a good fit to the actual time course the two fractions will always sum to approximately 1, which means that any variation in one fraction will be reflected in a complementary variation in the other. Thus, changes in, e.g. fL, with sample date or

location (as later discussed in 3.3.2) could be caused by change in the concentration of labile or semi-labile fraction, or both. If the focus is on the rate parameters, this is not so important, but if the magnitudes of the fractions are to be comparative metrics, the need to be independent of each other. For example, ft could be the actual component score, score normalized to t0 total fluorescence, or score normalized to DOC.

The authors offer the interpretation that the fractions correspond to "pools" of more or less reactive DOM. However, Murphy et al. (2018) interpret biexponential kinetics as possibly reflecting multiple photoreaction pathways, a fast one involving ROS and slower one related to direct photochemistry. How do the authors reconcile these differing interpretations?

---

## Referee Comment (RC2) · Anonymous Referee #2 · 10 Sep 2020

Overall comment: The study described in the manuscript aims to obtain a generalized method for studying light sensitivity of DOM in different water bodies. Authors suggest that this method may provide additional information on natural DOM quality. In general, I find this approach novel, however, some aspects are missing that would make me believe the method worse trying. The method seems to be very complicated and affected by numerous factors that the reader should keep in mind, but the discussion on the advantages of this method is very abstractive.

Specific comments: I have several more specific comments to the manuscript: Authors mentioned by themselves that pH changes might lead to reactions of flocculation, furthermore, pH may greatly affect EEM and absorbance signals which measured in this study, herewith I have a question, why did the Authors acidify the sample prior to ex-

tractions. And why did they use GFFs rather than 0.2 $\mu$m membrane filters? In my understanding, the colloidal fraction of DOM that is most likely to be present in the sample after 0.7$\mu$m filtration is the most susceptible to flocculation. The authors mentioned that natural samples were affected by cold storage. How exactly the sensitivity study was performed? The authors did a comparison study for a humic standard comparing extracted and not extracted solutions and saw a difference in the quality of DOM on EEM. Since quite altered by isolation and freeze-drying standard differ between extracting and non-extracting approach, the question is how would the natural samples differ? To which extent those extracts would be representative of what is actually going on in-situ? Also, the extracts were evaporated and diluted, what method was used for that? Microwave? The Authors describe precisely in details different approaches and drawbacks for the standard humic substance but the methods are lucking the description of the experiments performed on natural water samples. For instance, the reader learns that the natural organic matter was collected in different seasons only in the section "Results and discussion" and so on. Have the authors measured the initial conditions sample in each water bodies? How did EEM differ in terms of Fmax scores? Were they different? Also, for different seasons? I would expect that the Fmax scores would indicate that the composition of DOM is different between those water bodies, therefore I wonder whether the reported method would provide greater advantages than the outcome that DOM between water bodies is different? If the question is rather on the photosensitivity, in my opinion, some kind of discussion should be present on how comparable the results will be to what may happen in situ.

Technical corrections: Fig.9 Two plots for Comp3 For EEM comp. decay there are 4 curves for shallow and deep wetlands, were those replicates, or those different seasons measurements the authors have mentioned in the last chapter?

---

## Author Comment (AC1) · 1 Oct 2020

We appreciate the thoughtful and thorough comments, which we believe can be grouped into three substantial points – our use of a nitrite actinometer, our suggestions for improving reproducibility and comparability in photochemical degradation studies of DOM, and the novelty of our contribution in terms of its utility vs. existing approaches and our suggested interpretation of the data we generated.

We used a nitrite actinometer to illustrate the importance of controlling photon exposure and the potential limitations of modeling degradation processes as a function of time, shown distinctly in Figure 3. As the referee states, "Any photosensitivity study should include a detailed documentation of the optical setup and exposure measure-

ments such that the results can be expressed either as a function of time or cumulative exposure. Studies do range in how thoroughly exposure conditions are documented [emphasis ours]." We contend it is often difficult to calculate cumulative exposure from reported time alone in the published literature, making comparability difficult, and show that using a nitrite actinometer to calculate and report exposure directly is one way to improve this situation. We agree that other means of calculating and reporting cumulative exposure, such as measuring radiation across the UV spectrum directly, can only address this issue when irradiation design allows for accurate photon dose quantification with a spectroradiometer (e.g. irradiating quartz vessels/spectrophotometer cells with a known pathlength, cells are isolated from each other, ensuring that light is collimated/off axis photons do not hit samples, etc). Many labs have successfully designed systems that allow for very accurate photon dose quantification using solar simulators. For example, Powers and Miller (2015, MarChem, 171) irradiated 10 cm cells vertically in a black water-cooled aluminum block that maintains cells at a set temperature and allows no transfer of light between samples. Additionally, a gray 0.5" PVC plate with holes matching the spacing/interior diameter of the spectrophotometer cells was placed over the block and positioned under various 2" diameter Schott long-bandpass cutoff filters (280 nm to 480 nm) that were evenly spaced in a tray located at the bottom of the solar simulator exposure chamber. Due to the reflective exclusion of high angles at the surface of the Schott glass filters, reflectance at the face of the quartz cell is minimized. Furthermore, the gray plate reduces the entrance diameter and absorbs any off axis photons within the "gershun-like" tube formed by the gray plate, which also removed (or greatly minimized) reflectance at the face of the quartz cells by collimating the light. However, as mentioned in our manuscript, this type of experimental design allows for a limited number of samples, and therefore a limited number of measurements. Because the system described in Powers and Miller (2015) irradiates samples in 10 cm cells, even samples with relatively low CDOM values are not optically thin. While Powers and Miller (2015) used the methods of Hu et al. (2002) to correct "photons absorbed by CDOM, $Qa(\lambda)$" for shelf-shading (i.e. $Qa(\lambda) = E0(\lambda) \times S(1-\exp(-ag(\lambda) \times L) \times t$, where

E0($\lambda$) is the measured spectral irradiance entering each cell, S is the irradiated surface area, ag($\lambda$) is the CDOM absorption coefficient, L is the pathlength, and t is irradiation time), it is difficult to verify that these corrections are sufficient. It is possible that there are concentration effects but this is difficult to evaluate due to changes in sample matrix when diluting samples. Furthermore, samples in such a system must also be irradiated with no headspace because bubbles are terrible for optics, leading to oxygen loss during irradiations. While this may not be a problem for short exposures, many photoproducts of interest (i.e. $CO_2$) require long irradiations to generate a measurable product and it is possible that rates decrease at lower oxygen concentrations (e.g. Gao and Zepp, 1998). Lastly, this system allows for no pH control during irradiations, even though it is well known that pH can change during irradiations (Gao and Zepp, 1998) and photodegradation rates change at different pH values (Timko et al. 2015; Song et al. 2017). While some researchers have chosen to buffer samples to overcome this issue (e.g. Powers and Miller, 2015), to our knowledge it is still unknown how buffers may or may not impact photodegradation rates.

Therefore, our primary methodological objectives were to establish a reproducible design that would 1. Irradiate samples under "optically-thin" conditions, even at relatively high CDOM values. We therefore chose the quartz spiral flowcell with a 1 mm pathlength, which allows for 100x high CDOM values when compared to a 10 cm spec cell. One very surprising result was that even though SRNOM with DOC concentrations ranging from 10 to 50 mg L-1 degraded at similar rates, there appears to be a concentration dependence at low SRNOM DOC concentrations. This is a subject that we hope to evaluate more in future work. 2. To prevent oxygen loss, the sample is continually returned to an equilibrator, where it is mixed and re-oxygenated. 3. While we did not control samples for pH in this study (as mentioned no pH changes when irradiating our samples at pH 3), pH control is possible with a microtitrator and has been described in a previous study (Timko et al. 2015). We realize with this design we have sacrificed accurate photon dose quantification with a spectroradiometer that is possible in designs with very well defined and controlled optics. However, with the goal

of reproducibility in mind and limiting problems that occur when irradiating sealed vessels, we believe this system is a valuable tool to use when evaluating "photosensitivity". For instance, it will be a great tool to evaluate some of the questions/issues mentioned above (e.g. do buffers impact photodegradation rates?). Unfortunately, many studies do not control optics or do not use reproducible designs, e.g. irradiating samples in curved vessels on their sides or in flasks. While we cannot calculate spectral photon doses in our system, we thought that using nitrite actinometry as a broadband measurement of UVA photon dose was better to use than irradiation time as mentioned in the manuscript (lines 211-220, 259-268, Figure 3). We thank the reviewer for bringing up these valuable discussion points. A revised manuscript would discuss in detail the tradeoffs we made between radiometer-quantifiable optics and sample control.

The referee raises a few points related to our other suggestions for improving comparability and reproducibility when measuring photodegradation kinetics, stating "I quite agree with these recommendations which are straightforward to implement. For others, it is not clear that they will always contribute to reproducible, or more importantly, interpretable, results." Here we stress that we do not claim our approach provides a universally applicable prescription to ensure reproducible results – rather we hope to illustrate several obstacles to reproducing and comparing studies of photodegradation kinetics, and demonstrate some progress towards improving or at least identifying these sources of uncertainty. The manuscript text can be revised to convey this distinction more clearly. Specifically, the referee notes "[w]hile extracts are inherently more stable than raw filtrates, the authors demonstrate differences in the photosensitivity kinetics of extracts vs the original filtrates, the reasons for which are not well understood. The authors advise preparing extracts in some situations but not others, this leaves open the question of comparability."

This is in fact exactly our intention – to illustrate obstacles to reproducibility that need to be explicitly addressed when conducting or at least when comparing studies of photodegradation kinetics. As the referee notes, the differences between extracts and filtered raw water samples are not well understood and worthy of future study. Until these issues are resolved, researchers will have to weigh the advantages and disadvantages of extracts vs. raw water. Here we chose to use only extracts in subsequent trials in order to compare samples collected at different times that were extracted for stable storage, and to limit our inference to differences in photodegradation kinetics arising from DOM composition and not matrix differences. Other studies will have to choose between using extracts vs. raw water in accordance with study priorities. The differences we demonstrate (Figures 6 and 7, Table 1) illustrate underappreciated limitations to reproducibility and identify opportunities for further research into the effects of extraction on photodegradation kinetics and mechanisms of action (e.g. effect of matrix vs. fraction of DOM extracted). Along similar lines the referee notes "The authors also advise using a standard optical density in the sample, yet also show there is a concentration dependence to the kinetics, even for an optically thin exposure conditions. Again, a decision will have to made of whats more important, knowing how the DOM behaves at its natural concentration or being able to compare it to other sources." Again, we are in agreement with the referee here, and believe one of our manuscript's contributions is highlighting these underappreciated sources of variability in DOM photodegradation kinetics. This issue in particular is vexing, as differences between DOM concentrations under conditions meeting conventional thresholds for optically thin solutions are not explained well by currently published literature. We will revise the manuscript text to make it more clear that our goals were not to show only reproducible results but to identify underappreciated sources of variability that need to be controlled, and to use these insights to conduct limited trials with comparable results.

The referee's final point concerns the utility of our approach in a wider sense – "In the end, investigators will need more information to decide whether the additional insight gained from defining photosensitivity kinetics will be worth the efforts needed to perform all these standardization steps. This report documents experimental tests that show what are sources of variability in the procedure but does not show what new understanding is actually being gained from the results (vs what can already be gained

from other optical/chemical measurements typically made on DOM)." We agree that our approach is not trivial, and some applications may not really benefit from this level of effort. Our primary research goal was to compare the sensitivity of different DOM sources to photochemical degradation processes at environmentally relevant temporal scales. Measuring kinetics of DOM photodegradation changes seemed the best way to pursue this, but we found existing approaches to describing DOM photodegradation kinetics allowed for possible uncertainties that were barriers to meaningful comparison (as discussed in detail above). We believe the methodological suggestions we present are useful to other researchers in this area pursuing similar goals, and our research findings (seasonal and spatial differences in wetland DOM photo-sensitivity, with impli-cations for understanding DOM composition and the other ecological processes that affect it in this setting) demonstrates the potential utility of this approach and illustrates phenomena worthy of further investigation.

Specific concerns (referee's comments bracketed by \*\*\*):

\*\*\*Line 140 "Solar exposure" – This was not solar exposure, but a laboratory set up. Better (?) "Total sample exposure"\*\*\*

We agree and will revise the manuscript accordingly

\*\*\*Figure 7 and Table 1 – The captions for these describe kL and kSL with units of mol photons m-2. Those are the units of P, the units of k should be the inverse, m2 [mol photons]-1.\*\*\*

We agree and will revise the manuscript accordingly.

\*\*\*Figure 7 and Table 1 apparently show the same data – and both state the statistics of the comparison. Questionable whether both are necessary. Figure 7 caption, however, says see Fig. 5 for data. It appears that Fig. 6 has the data. \*\*\*

The referee is correct that Figure 7 caption should refer to Figure 6 instead of Figure 5, and also correct that both Figure 7 and Table 1 show the same data. Readers may find

the graphical comparison easier to quickly parse, but the tabular presentation may be more useful to future studies seeking comparison. We think the graphical presentation is more valuable to readers in the main text, and would move Table 1 to an appendix upon revision.

***Line 350-360 – Factors affecting storage of filtered water. Previous text only mentioned preparation of filtrates using 0.7 _M glass fiber filters, which allows passage of a substantial bacterial fraction. However, subsequent guidelines caution against used stored material even filtered to 0.2 _M (some bacteria can even get through those). Please clarify what filtration method was used for the stored sample, but in any case (slow) microbial degradation may be another factor resulting in changes in stored material.***

We inadvertently failed to note an important step in our methods – all samples, whether raw water or solid-phase extracts redissolved in water, were filtered through syringe-mounted 0.2 $\mu$m cellulose acetate filters that were pre-rinsed with > 30 mL ultrapure C-free water. GFF filters were used as a pre-filter immediately after samples were returned to the lab, before either short-term storage or solid-phase extraction. Furthermore, since 0.7 $\mu$m filters were combusted, their true pore size is probably smaller than 0.7 $\mu$m (0.3 $\mu$m in Nayar and Chou, 2003) and therefore be comparable 0.2 $\mu$m filters (Nayar and Chou, 2003). We apologize for the omission and ensuing confusion. The referee notes that microbial degradation is still possible even when using a filter of 0.2 $\mu$m pore sizes – we agree (see Brailsford et al., 2017; Luef et al., 2015), and believe this supports our recommendation to either use extracts or keep storage time minimal and highlights the need for more research in this area. We also believe microbial degradation/contamination is minimal during exposures under the solar simulator using optically thin conditions (e.g. Stubbins et al. 2017, 122), but can revise our manuscript to mention the full range of possibilities that must be accounted for in any similar experimental set up. This would include discussion of between-sample cleaning with dilute NaOH and occasional flushes of sample lines with isopropanol (followed by extensive

water flushes and testing for DOC residue) to prevent microbial contamination of flow lines.

***Figure 10 – Caption says see Figure 6 for data, data appears to be in Fig. 9. This correction also applies to the caption for Figure A1. Figure 10 could be misleading to the reader, as a quick inspection may suggest that the spread of the multiple points for a particular sample source (and standard deviation in the table) are indications of reproducibility. Actually, only the SRNOM and three of the small wetland points are repeat determinations on the same sample. Separating the points for different sample dates would avoid this confusion, especially given that later on (section 3.3.2) the authors attribute some of the difference in the wetland results with time to possible differences in sample composition and/or previous solar exposure.***

The referee is correct that the captions for Figure 10 and A1 should read "see Fig. 6 for data". We agree that the representation of data in Figure 10 may be confusing. This criticism may also apply to Table 2, where group means and standard deviations for photodegradations of DOM sources with more than one trial are grouped differently – SRNOM represents three actual replicate trials using the same sample source, the three "large wetland" trials represent different samples (collected at same site but at different times), and the five "small wetland" trials represent three replicate trials using the same source and two trials from different sample sources (collected at the same site but at different times). We originally thought it was useful to make the comparison in Figure 10 in this way because of the widespread interest in differentiating DOM sources by setting. By showing fitted parameters as independent data points we thought readers could parse the different types of grouping by reading the text carefully. However, we agree with the referee's concerns – at the very least the sampling dates should be differentiated by using different symbols in the plot. Upon reflection we also believe Table 2 should be changed along these lines – our original intention was to facilitate comparisons between environmental settings, but calculating group means with these data is not appropriate given their different structures. Means for replicated trials are

sensible, and it may even be valuable to see how un-replicated trials with samples from different seasons at a site differ, as solely spatial comparisons often do not account for temporal/seasonal variation but readers may still be interested in summarizing data by site. But calculating one mean from data that includes 3 replicated trials of the same sample source and two further distinct samples does not convey useful information. We think publishing tabular data may be useful for future research, but this data would be better represented separately. For size reasons, such a table may be better suited to an appendix.

***Line 456ff, section 3.3.1 Here the relative abundances of kinetic components are considered as metrics of DOM composition or other influences on photosensitivity. A problemwith the comparative use of fL and fSL is that the two parameters are not independent. Assuming a good fit to the actual time course the two fractions will always sumto approximately 1, which means that any variation in one fraction will be reflected in a complementary variation in the other. Thus, changes in, e.g. fL, with sample date or location (as later discussed in 3.3.2) could be caused by change in the concentration of labile or semi-labile fraction, or both. If the focus is on the rate parameters, this is not so important, but if the magnitudes of the fractions are to be comparative metrics, the need to be independent of each other. For example, ft could be the actual component score, score normalized to t0 total fluorescence, or score normalized to DOC.***

We see the referee's point that fSL and fL are not truly independent. They were fitted as separate coefficients in the nonlinear fitting algorithm applied to the data, but should indeed sum to 1, so either could be expressed as the difference between 1 and its counterpart (i.e. fSL = 1- fL or vice versa). We expressed them separately here in accordance with their separate representation in the equation conventionally used to define the decay model (Equation 2, which Equation 3 then followed), and because they were fitted separately (which would make the choice of which to use to find its counterpart arbitrary). We agree that this causes ambiguities when trying to understand differences in these values between samples, as in the example offered by the referee.

We do believe using these values may be informative, as a major part of our claim to novelty rests on pointing out possible connections between these values and environmental phenomena (such as previous exposure to photodegradation or differences in source material composition) instead of basing inference solely on k parameters. We are not sure how to best resolve this issue. ft already represents PARAFAC component fluorescence normalized to time 0 for each sample. Further processing the data to derive a single value (like the ratio of fSL and fL) adds another layer of abstraction to a situation that already involves multiple modeling steps, and seems harder to interpret in a way that materially connects to the phenomena described. A revised manuscript would discuss these points. However, We still think it is valuable to compare fL between samples – even if differences in this value represent multiple possible causes as described by the referee, differences here represent real observed differences in photodegradation behavior. Normalization to [DOC] or to absorbance at some reference wavelength may be useful in this context. We believe this is an area ripe for future research, and view our contribution in part as pointing out the need for attention to previously unexplored dimensions of current paradigms of DOM photodegradation.

***The authors offer the interpretation that the fractions correspond to "pools" of more or less reactive DOM. However, Murphy et al. (2018) interpret biexponential kinetics as possibly reflecting multiple photoreaction pathways, a fast one involving ROS and slower one related to direct photochemistry. How do the authors reconcile these differing interpretations?***

We appreciate the referee's thoughtful reading of our conclusions. Our conceptualization – differences in photodegradation kinetics arising from different relative quantities of "pools" reacting at relatively fast and slow rates – may be unnecessarily general or vague. However, we think the explanation from Murphy et al. (2018) invoked by the referee is not necessarily opposed to the "pools" we ascribe to the model. We consider these "pools" in a general sense. Given the immense complexity of natural DOM, it is unlikely there are two distinct, non-overlapping compositional types of photo-sensitive

DOM. Yet the excellent fit of biexponential models across diverse DOM sources requires some explanation. What we refer to generally as reactive pools could more specifically be the pools of DOM subject to the respective reaction pathways suggested by Murphy et al. and their associated rate constants. Or perhaps it is more appropriate to consider a single "pool" with differing capacities for two classes of reactions. We are not able to definitively conclude what lies behind the biexponential model – but believe our novel attention to the f parameters in the model equation represents the kind of exploration that is needed to make progress in this direction.

References

Brailsford, F. L., Glanville, H. C., Marshall, M. R., Golyshin, P. N., Johnes, P. J., Yates, C. A., Owen, A. T., & Jones, D. L. (2017). Microbial use of low molecular weight DOM in filtered and unfiltered freshwater: Role of ultra-small microorganisms and implications for water quality monitoring. Science of The Total Environment, 598, 377–384. https://doi.org/10.1016/j.scitotenv.2017.04.049

Gao, H., & Zepp, R. G. (1998). Factors Influencing Photoreactions of Dissolved Organic Matter in a Coastal River of the Southeastern United States. Environmental Science & Technology, 32(19), 2940–2946. https://doi.org/10.1021/es9803660

Hu, C., Muller-Karger, F. E., & Zepp, R. G. (2002). Absorbance, absorption coefficient, and apparent quantum yield: A comment on common ambiguity in the use of these optical concepts. Limnology and Oceanography, 47(4), 1261–1267. https://doi.org/10.4319/lo.2002.47.4.1261

Luef, B., Frischkorn, K. R., Wrighton, K. C., Holman, H.-Y. N., Birarda, G., Thomas, B. C., Singh, A., Williams, K. H., Siegerist, C. E., Tringe, S. G., Downing, K. H., Comolli, L. R., & Banfield, J. F. (2015). Diverse uncultivated ultra-small bacterial cells in groundwater. Nature Communications, 6(1), 6372. https://doi.org/10.1038/ncomms7372

Murphy, K. R., Timko, S. A., Gonsior, M., Powers, L. C., Wünsch, U. J., & Stedmon, C. A. (2018). Photochemistry Illuminates Ubiquitous Organic Matter Fluorescence Spectra. Environmental Science & Technology, 52(19), 11243–11250. https://doi.org/10.1021/acs.est.8b02648

Nayar, S., & Chou, L. M. (2003). Relative efficiencies of different filters in retaining phytoplankton for pigment and productivity studies. Estuarine, Coastal and Shelf Science, 58(2), 241–248. https://doi.org/10.1016/S0272-7714(03)00075-1

Powers, L. C., & Miller, W. L. (2015). Photochemical production of CO and CO2 in the Northern Gulf of Mexico: Estimates and challenges for quantifying the impact of photochemistry on carbon cycles. Marine Chemistry, 171, 21–35. https://doi.org/10.1016/j.marchem.2015.02.004

Song, G., Li, Y., Hu, S., Li, G., Zhao, R., Sun, X., & Xie, H. (2017). Photobleaching of chromophoric dissolved organic matter (CDOM) in the Yangtze River estuary: Kinetics and effects of temperature, pH, and salinity. Environmental Science: Processes & Impacts, 19(6), 861–873. https://doi.org/10.1039/C6EM00682E

Stubbins, A., Mann, P. J., Powers, L., Bittar, T. B., Dittmar, T., McIntyre, C. P., Eglinton, T. I., Zimov, N., & Spencer, R. G. M. (2017). Low photolability of yedoma permafrost dissolved organic carbon. Journal of Geophysical Research: Biogeosciences, 122(1), 200–211. https://doi.org/10.1002/2016JG003688

Timko, S. A., Gonsior, M., & Cooper, W. J. (2015). Influence of pH on fluorescent dissolved organic matter photo-degradation. Water Research, 85, 266–274. https://doi.org/10.1016/j.watres.2015.08.047

---

## Author Comment (AC2) · 1 Oct 2020

We appreciate the thoughtful comments from the referee. The most pressing issue presented in this review is how to justify the effort involved. Our main goal was to compare the sensitivity of different DOM sources to photochemical alteration to better understand potential influences on DOM composition in natural waters. There are many approaches to characterizing the optical properties different DOM sources and an established tradition of photochemical experimentation on DOM. However, DOM in natural waters is subject to many potential transformation processes that may compete and interact on different temporal scales, so reliable comparison of degradation kinetics seemed like the most promising approach to characterize photosensitivity under environmentally relevant photon exposures. For example, changes to DOM composition that slowly over the equivalent of several days of photodegradation may be less important than changes occurring in a few hours in an environment where water is moving or mixing and subject to other processes like microbial transformations. Much of the current manuscript is devoted to the methodological considerations we encountered in our effort to reliably compare degradation kinetics, many of which are also relevant to other approaches used to study DOM photodegradation and thus broadly interesting to this community. Our specific research findings were that wetland DOM photo-sensitivity varied in space and time with potential implications for understanding DOM composition in this setting generally, and that our method identified particular aspects of photodegradation kinetics that might represent links between DOM composition, environmental phenomena, and photodegradation. This represents an initial foray into understanding the role of photodegradation in shaping DOM composition in our study area, and we hope it illustrates aspects of DOM photodegradation that may be applicable elsewhere and interest the broader research community.

The referee raised several methodological questions. We acidified our samples to pH 3 because we had to control pH - pH changes produce fluorescence changes that confound the fluorescence decay due to photodegradation reactions under investigation. We did not wish to use chemical buffers, as some of these may also interfere with fluorescence and may have unknown effects on degradation kinetics. Active control of pH during experiments using autotitration is another option we did not use because the electrode contaminated samples during prolonged irradiations. Therefore, we needed to be sure that pH would be consistent between samples and during photodegradation of each sample. Starting samples at pH 3 allowed the full protonation of most organic acids (Ritchie and Perdue, 2003), which should prevent solution pH change due to the photoproduction of $CO_2$. We recognize that this itself presents an obstacle to reproducibility or comparison with other studies, as kinetics are greatly affected by pH, but this seemed the most reliable way to avoid pH change during photodegradation to allow comparisons between our samples. Another methodological question involved filtration choices. We appreciate the careful attention of the reviewer here,

which helped us discover that we inadvertently failed to note an important step in our methods – all samples, whether raw water or solid-phase extracts redissolved in water, were filtered through syringe-mounted 0.2 $\mu$m cellulose acetate filters that were pre-rinsed with > 30 mL ultrapure C-free water. GFF filters were used as a pre-filter immediately after samples were returned to the lab, before either short-term storage or solid-phase extraction. We apologize for the omission and ensuing confusion. Additionally, as mentioned to reviewer 1, because 0.7 $\mu$m filters were combusted, their true pore size is probably smaller than 0.7 $\mu$m and likely be comparable 0.2 $\mu$m filters (Nayar and Chou, 2003).

The referee's methodological questions included several about comparability of our results to natural, un-processed DOM. These included questions about effects of extraction, the means used during evaporation of methanol extracts, details on the experiment showing effects of storage, use of freeze-dried IHSS standards, and questions about the collection and initial characterization of natural DOM sources. A few of these questions were addressed in the text The methanol extracts were evaporated in an open vial under a fume hood under a stream of high-purity N2 gas – the gas flow created turbulence in methanol, facilitating its evaporation. We used IHSS SRNOM standard as the DOM source in many of our experiments because it has been extensively studied and provides a benchmark against which other photodegradation kinetics experiments can compare results. Reconstituted freeze-dried SRNOM and its filtered source water showed very similar molar absorptivity at 280 nm and fluorescence index values when collected (Kuhn et al., 2014). While this may not translate to identical kinetics of photodegradation, it suggests optical properties are not radically transformed during freeze-drying and reconstitution. We believe the advantages of using a standardized material in wide circulation outweigh any artifact of freeze-drying. The experiment showing the effects of storage on photodegradation kinetics was conducted according to different protocols than following experiments – samples had higher DOC concentrations, and a different flow cell with shorter path length was used. This experiment was the first conducted, chronologically, of those presented

here, and its failures of replication motivated further work and helped us develop our protocols. We apologize if introducing these protocols sowed confusion, and will edit the manuscript to ensure it is clear which protocols applied to which experiments. The referee also asked whether EEMs and other measures were available for comparison between the solid-phase extracts used throughout the paper and their original raw water DOM sources. While these are not available (at least in comparable form) for many of the samples used to show the range of variability across a wide gradient of DOM sources (e.g. the stream, coastal ocean, and Sargassum sp.), we did collect EEMs and absorbance spectra for the "large" and "small" wetlands sampled in different seasons at the time of collection. We will revise the manuscript to include comparisons between source water EEMs and extract EEMs where possible.

The referee ended their specific comments by requesting "some kind of discussion. . .on how comparable the results [from extracts studied in the lab] will be to what may happen in situ." We appreciate the utility of this kind of information – inference on the processes at work in the natural setting is the original motivation for our entire effort. While we were able to determine that extracts behaved differently than raw water (demonstrated with SRNOM), we cannot identify the mechanistic basis of these differences with the current evidence. The difference between in situ conditions and our lab results might be useful to expound further in a future manuscript, however, this is not a trivial task. For instance, we used solid phase extracts to minimize matrix effects and controlled pH to look at differences in photochemistry between DOM samples and SRNOM. Cations and in particular magnesium, may actually enhance DOM fluorescence signals (Stichak et al. 2019) and pH also influences sample optical properties, as mentioned previously. Metals like iron are known to enhance absorbance measurements and potentially quench fluorescence (Poulin et al. 2014). Halides may also enhance photodegradation rates (Grebel et al. 2009). These potential matrix effects (see Li and Hur, 2017 for a detailed discussion) make observed differences in whole water optical properties difficult to interpret. SPE allows for comparisons with minimal or entirely without potential interferences and matrix effects. While sample

matrix is indeed different in situ, our method may best be described as measuring potential photodegradation of DOM itself, which is why we describe it as photosensitivity. One of our key results shows the utility of our method in this regard – we are able to measure the relative differences in fast degradation processes (described by fL and kL), which may be more environmentally relevant than the slower processes in the context of constantly changing DOM subject to a panoply of other transformation processes and constantly in motion. We thank the referee for asking for this kind of discussion, and will revise the manuscript to make the relationships between our lab approach and in situ inferences more clear.

The referee's final question was about Figure 9. There is an error in this figure that may have produced additional confusion – the left plot is of PARAFAC component 3 while the right plot is of PARAFAC component 4. We will correct this in the manuscript. The referee's question concerned the provenance of the multiple curves shown for the wetland samples. These plots show the full data available for all different DOM sources subjected to the final protocol. This includes the three replicates of extracts from the same "small wetland" DOM source and two trials with extracts from "small wetland" DOM collected at different times, three trials with extracts from "large wetland" DOM collected at different times, and the three replicates of SRNOM extracts. As described in our response to RC1, we recognize the need to clarify the different dimensions of these multiple trials for each site/source, especially in Figure 10 and Table 2. It may be useful to further clarify here as well, though as this figure shows rawer data (PARAFAC model component fluorescence change over time) rather than biexponential model parameters for each curve shown here, this may be less pressing for this figure.

References

Grebel, J. E., Pignatello, J. J., Song, W., Cooper, W. J., & Mitch, W. A. (2009). Impact of halides on the photobleaching of dissolved organic matter. Marine Chemistry, 115(1), 134–144. https://doi.org/10.1016/j.marchem.2009.07.009

Kuhn, K. M., Neubauer, E., Hofmann, T., von der Kammer, F., Aiken, G. R., & Maurice, P. A. (2014). Concentrations and Distributions of Metals Associated with Dissolved Organic Matter from the Suwannee River (GA, USA). Environmental Engineering Science, 32(1), 54–65. https://doi.org/10.1089/ees.2014.0298

Li, P., & Hur, J. (2017). Utilization of UV-Vis spectroscopy and related data analyses for dissolved organic matter (DOM) studies: A review. Critical Reviews in Environmental Science and Technology, 47(3), 131–154. https://doi.org/10.1080/10643389.2017.1309186

Nayar, S., & Chou, L. M. (2003). Relative efficiencies of different filters in retaining phytoplankton for pigment and productivity studies. Estuarine, Coastal and Shelf Science, 58(2), 241–248. https://doi.org/10.1016/S0272-7714(03)00075-1

Poulin, B. A., Ryan, J. N., & Aiken, G. R. (2014). Effects of Iron on Optical Properties of Dissolved Organic Matter. Environmental Science & Technology, 48(17), 10098–10106. https://doi.org/10.1021/es502670r

Ritchie, J. D., & Perdue, E. M. (2003). Proton-binding study of standard and reference fulvic acids, humic acids, and natural organic matter. Geochimica et Cosmochimica Acta, 67(1), 85–96. https://doi.org/10.1016/S0016-7037(02)01044-X

T. Stirchak, L., J. Moor, K., McNeill, K., & James Donaldson, D. (2019). Differences in photochemistry between seawater and freshwater for two natural organic matter samples. Environmental Science: Processes & Impacts, 21(1), 28–39. https://doi.org/10.1039/C8EM00431E

---

## Author Response (AR1)

General Comments

Measurements of spectral fluoresence have become a standard part of any study of the dynamics and distribution of dissolved organic matter in both freshwater and marine environments. Methods for deriving standardized metrics from fluorescence data are well established, and most studies of DOM dynamics acquire detailed sets of excitation-emission matrices and analyze them using parallel factor analysis (PARAFAC). In the present contribution, Armstrong, et al., argue that the kinetics of fluorescence photo-sensitivity are also useful properties to characterize DOM, in particular, the magnitude and rate constants of biexponential kinetics. However, several important considerations

need to be observed if these photosensitivity metrics are to be compared across studies. These are shown in the context of a specialized continuous loop exposure/assay system that the authors have devised, but several are applicable to any photosensitivity study. One is the use of a reference material, SRNOM, to both test the derivation of PARAFAC components and standardize the kinetics for a given exposure set up. Another informative test is the reproducibility of the results in repeated runs including those by different observers, which helps to establish the uncertainty in the estimated kinetic parameters.

I quite agree with these recommendations which are straightforward to implement. For others, it is not clear that they will always contribute to reproducible, or more importantly, interpretable, results. One is the preparation of solid phase extracts of DOM vs using fresh filtrates for the measurements. While extracts are inherently more stable than raw filtrates, the authors demonstrate differences in the photosensitivity kinetics of extracts vs the original filtrates, the reasons for which are not well understood. The authors advise preparing extracts in some situations but not others, this leaves open the question of comparability. The authors also advise using a standard optical density in the sample, yet also show there is a concentration dependence to the kinetics, even for an optically thin exposure conditions. Again, a decision will have to made of whats more important, knowing how the DOM behaves at its natural concentration or being able to compare it to other sources.

The authors also advocate expressing kinetic results as a function of cumulative photon exposure as opposed to time, this will adjust for differences in exposure regime between different studies. In principle, this is a good idea. Any photosensitivity study should include a detailed documentation of the optical setup and exposure measurements such that the results can be expressed either as a function of time or cumulative exposure. Studies do range in how thoroughly exposure conditions are documented. For these kinetic studies, exposure has been measured with a nitrite actinometer. Actinometry is very useful to quantitate the effects within a specific optical set up and is

used effectively by the authors to monitor exposure between experiments, including those with variable working volume. However, the nitrite actinometer only measures quantum fluxes between 330 and 380 nm. For incident solar UV, this is only about 60 per cent of the spectrum (by quanta). For comparison across all photosensitivity studies, I would suggest a more general exposure metric, e.g. total UV radiation. Given the broadband 330-380 nm quantum flux, total UV can be estimated with reference to the manufacturer's stated spectral distribution. But lamp and component aging can change that (even adjusting for constant power), so the best approach is to measure the spectrum in conjunction with actinometry (admittedly, neither of these are trivial to perform).

In the end, investigators will need more information to decide whether the additional insight gained from defining photosensitivity kinetics will be worth the efforts needed to perform all these standardization steps. This report documents experimental tests that show what are sources of variability in the procedure but does not show what new understanding is actually being gained from the results (vs what can already be gained from other optical/chemical measurements typically made on DOM). As far as reproducibility, there is only one instance of repeat determination on a "real world" sample, the small wetland. Further repeat determinations are needed to demonstrate the general reproducibility of the approach. Also, some modification of how the photosensitive fractions are estimated may be needed if the results are to be used in a comparative context (see specific comments below). Despite that this is still a work in progress, it is quite appropriate to have the authors' study and recommendations in a Biogeosciences discussion article. Comments by a wide range of readers should help refine the ideas proposed and ultimately advance the field.

Specific comments:

Line 140 "Solar exposure" – This was not solar exposure, but a laboratory set up. Better (?) "Total sample exposure"

[Figure]

Figure 7 and Table 1 – The captions for these describe kL and kSL with units of mol photons m$^{-2}$. Those are the units of P, the units of k should be the inverse, m$^2$ [mol photons]$^{-1}$.

Figure 7 and Table 1 apparently show the same data – and both state the statistics of the comparison. Questionable whether both are necessary. Figure 7 caption, however, says see Fig. 5 for data. It appears that Fig. 6 has the data.

Line 350-360 – Factors affecting storage of filtered water. Previous text only mentioned preparation of filtrates using 0.7 $\mu$M glass fiber filters, which allows passage of a substantial bacterial fraction. However, subsequent guidelines caution against used stored material even filtered to 0.2 $\mu$M (some bacteria can even get through those). Please clarify what filtration method was used for the stored sample, but in any case (slow) microbial degradation may be another factor resulting in changes in stored material.

Figure 10 – Caption says see Figure 6 for data, data appears to be in Fig. 9. This correction also applies to the caption for Figure A1. Figure 10 could be misleading to the reader, as a quick inspection may suggest that the spread of the multiple points for a particular sample source (and standard deviation in the table) are indications of reproducibility. Actually, only the SRNOM and three of the small wetland points are repeat determinations on the same sample. Separating the points for different sample dates would avoid this confusion, especially given that later on (section 3.3.2) the authors attribute some of the difference in the wetland results with time to possible differences in sample composition and/or previous solar exposure.

Line 456ff, section 3.3.1 Here the relative abundances of kinetic components are considered as metrics of DOM composition or other influences on photosensitivity. A problem with the comparative use of fL and fSL is that the two parameters are not independent. Assuming a good fit to the actual time course the two fractions will always sum to approximately 1, which means that any variation in one fraction will be reflected in a complementary variation in the other. Thus, changes in, e.g. fL, with sample date or

location (as later discussed in 3.3.2) could be caused by change in the concentration of labile or semi-labile fraction, or both. If the focus is on the rate parameters, this is not so important, but if the magnitudes of the fractions are to be comparative metrics, the need to be independent of each other. For example, ft could be the actual component score, score normalized to t0 total fluorescence, or score normalized to DOC.

The authors offer the interpretation that the fractions correspond to "pools" of more or less reactive DOM. However, Murphy et al. (2018) interpret biexponential kinetics as possibly reflecting multiple photoreaction pathways, a fast one involving ROS and slower one related to direct photochemistry. How do the authors reconcile these differing interpretations?

[Figure]

Biogeosciences Discuss.,
https://doi.org/10.5194/bg-2020-207-RC2, 2020

[Figure]

Overall comment: The study described in the manuscript aims to obtain a generalized method for studying light sensitivity of DOM in different water bodies. Authors suggest that this method may provide additional information on natural DOM quality. In general, I find this approach novel, however, some aspects are missing that would make me believe the method worse trying. The method seems to be very complicated and affected by numerous factors that the reader should keep in mind, but the discussion on the advantages of this method is very abstractive.

Specific comments: I have several more specific comments to the manuscript: Authors mentioned by themselves that pH changes might lead to reactions of flocculation, furthermore, pH may greatly affect EEM and absorbance signals which measured in this study, herewith I have a question, why did the Authors acidify the sample prior to ex-

tractions. And why did they use GFFs rather than 0.2 $\mu$m membrane filters? In my understanding, the colloidal fraction of DOM that is most likely to be present in the sample after 0.7$\mu$m filtration is the most susceptible to flocculation. The authors mentioned that natural samples were affected by cold storage. How exactly the sensitivity study was performed? The authors did a comparison study for a humic standard comparing extracted and not extracted solutions and saw a difference in the quality of DOM on EEM. Since quite altered by isolation and freeze-drying standard differ between extracting and non-extracting approach, the question is how would the natural samples differ? To which extent those extracts would be representative of what is actually going on in-situ? Also, the extracts were evaporated and diluted, what method was used for that? Microwave? The Authors describe precisely in details different approaches and drawbacks for the standard humic substance but the methods are lucking the description of the experiments performed on natural water samples. For instance, the reader learns that the natural organic matter was collected in different seasons only in the section "Results and discussion" and so on. Have the authors measured the initial conditions sample in each water bodies? How did EEM differ in terms of Fmax scores? Were they different? Also, for different seasons? I would expect that the Fmax scores would indicate that the composition of DOM is different between those water bodies, therefore I wonder whether the reported method would provide greater advantages than the outcome that DOM between water bodies is different? If the question is rather on the photosensitivity, in my opinion, some kind of discussion should be present on how comparable the results will be to what may happen in situ.

Technical corrections: Fig.9 Two plots for Comp3 For EEM comp. decay there are 4 curves for shallow and deep wetlands, were those replicates, or those different seasons measurements the authors have mentioned in the last chapter?

—————————————

[Figure]

**RC1 response**
We appreciate the thoughtful and thorough comments, which we believe raise three substantial points – justifying our use of a nitrite actinometer, our suggestions for improving reproducibility and comparability in photochemical degradation studies of DOM, and the novelty of our contribution in terms of its utility vs. existing approaches and our suggested interpretation of the data we generated.

We used a nitrite actinometer to illustrate the importance of controlling photon exposure and the potential limitations of modeling degradation processes as a function of time, shown distinctly in Figure 3. As the referee states, "Any photosensitivity study should include a detailed documentation of the optical setup and exposure measurements *such that the results can be expressed either as a function of time or cumulative exposure*. Studies do range in how thoroughly exposure conditions are documented [emphasis ours]." We contend it is often difficult to calculate cumulative exposure from reported time alone in the published literature, making comparability difficult, and show that using a nitrite actinometer to calculate and report exposure directly is one way to improve this situation. We agree that other means of calculating and reporting cumulative exposure, such as measuring radiation across the UV spectrum directly, can only address this issue when irradiation design allows for accurate photon dose quantification with a spectroradiometer (e.g. irradiating quartz vessels/spectrophotometer cells with a known pathlength, cells are isolated from each other, ensuring that light is collimated/off axis photons do not hit samples, etc). Many labs have successfully designed systems that allow for very accurate photon dose quantification using solar simulators. For example, Powers and Miller (2015, MarChem, 171) irradiated 10 cm cells vertically in a black water-cooled aluminum block that maintains cells at a set temperature and allows no transfer of light between samples. Additionally, a gray 0.5" PVC plate with holes matching the spacing/interior diameter of the spectrophotometer cells was placed over the block and positioned under various 2" diameter Schott long-bandpass cutoff filters (280 nm to 480 nm) that were evenly spaced in a tray located at the bottom of the solar simulator exposure chamber. Due to the reflective exclusion of high angles at the surface of the Schott glass filters, reflectance at the face of the quartz cell is minimized. Furthermore, the gray plate reduces the entrance diameter and absorbs any off axis photons within the "gershun-like" tube formed by the gray plate, which also removed (or greatly minimized) reflectance at the face of the quartz cells by collimating the light. However, as mentioned in our manuscript, this type of experimental design allows for a limited number of samples, and therefore a limited number of measurements. Because the system described in Powers and Miller (2015) irradiates samples in 10 cm cells, even samples with relatively low CDOM values are not optically thin. While Powers and Miller (2015) used the methods of Hu et al. (2002) to correct "photons absorbed by CDOM, $Q_a(\lambda)$" for shelf-shading (i.e. $Q_a(\lambda) = E_0(\lambda) \times S(1-\exp(-a_g(\lambda) \times L) \times t$, where $E_0(\lambda)$ is the measured spectral irradiance entering each cell, S is the irradiated surface area, $a_g(\lambda)$ is the CDOM absorption coefficient, L is the pathlength, and t is irradiation time), it is difficult to verify that these corrections are sufficient. It is possible that there are concentration effects but this is difficult to evaluate due to changes in sample matrix when diluting samples. Furthermore, samples in such a system must also be irradiated with no headspace because bubbles are terrible for optics, leading to oxygen loss during irradiations. While this may not be a problem for short exposures, many photoproducts of interest (i.e. $CO_2$) require long irradiations to generate a measurable product and it is possible that rates decrease at lower oxygen concentrations (e.g. Gao and Zepp, 1998, ES&T, 32). Lastly, this system allows for no pH control during irradiations, even though it is well known that pH can change during irradiations (Gao and Zepp, 1998) and photodegradation rates change at different pH values (Timko et al. 2015, Water Res.; Song et al. 2017, ES:P&I, 19). While some researchers have chosen to buffer samples to overcome this issue (e.g. Powers and Miller, 2015), to our knowledge it is still unknown how buffers may or may not impact photodegradation rates.

Therefore, our primary methodological objectives were to establish a reproducible design that would 1. Irradiate samples under "optically-thin" conditions, even at relatively high CDOM values. We therefore chose the quartz spiral flowcell with a 1 mm pathlength, which allows for 100x high CDOM values when compared to a 10 cm spec cell. One very surprising result was that even though SRNOM with DOC concentrations ranging from 10 to 50 mg $L^{-1}$ degraded at similar rates, there appears to be a concentration dependence at low SRNOM DOC concentrations. This is a subject that we hope to evaluate more in future work. 2. To prevent oxygen loss, the sample is continually returned to an equilibrator, where it is mixed and re-oxygenated. 3. While we did not control samples for pH in this study (as mentioned no pH changes when irradiating our samples at pH 3), pH control is possible with a microtitrator and has been described in a previous study (Timko et al. 2015). We realize with this design we have sacrificed accurate photon dose quantification with a spectroradiometer that is possible in designs with very well defined and controlled optics. However, with the goal of reproducibility in mind and limiting problems that occur when irradiating sealed vessels, we believe this system is a valuable tool to use when evaluating "photosensitivity". For instance, it will be a great tool to evaluate some of the questions/issues mentioned above (e.g. do buffers impact photodegradation rates?). Unfortunately, many studies do not control optics or do not use reproducible designs, e.g. irradiating samples in curved vessels on their sides or in flasks. While we cannot calculate spectral photon doses in our system, we thought that using nitrite actinometry as a broadband measurement of UVA photon dose was better to use than irradiation time as mentioned in the manuscript (lines 211-220, 259-268, Figure 3). We thank the reviewer for bringing up these valuable discussion points. Revisions discuss in detail the tradeoffs we made between radiometer-quantifiable optics and sample control (lines 424-430).

The referee raises a few points related to our other suggestions for improving comparability and reproducibility when measuring photodegradation kinetics, stating "I quite agree with these recommendations which are straightforward to implement. For others, it is not clear that they will always contribute to reproducible, or more importantly, interpretable, results." Here we stress that we do not claim our approach provides a universally applicable prescription to ensure reproducible results – rather we hope to illustrate several obstacles to reproducing and comparing studies of photodegradation kinetics, and demonstrate some progress towards improving or at least identifying these sources of uncertainty. The manuscript was revised to convey this distinction more clearly (lines 71-72; 576-578). Specifically, the referee notes "[w]hile extracts are inherently more stable than raw filtrates, the authors demonstrate differences in the photosensitivity kinetics of extracts vs the original filtrates, the reasons for which are not well understood. The authors advise preparing extracts in some situations but not others, this leaves open the question of comparability."

This is in fact exactly our intention – to illustrate obstacles to reproducibility that need to be explicitly addressed when conducting or at least when comparing studies of photodegradation kinetics. As the referee notes, the differences between extracts and filtered raw water samples are not well understood and worthy of future study. Until these issues are resolved, researchers will have to weigh the advantages and disadvantages of extracts vs. raw water. Here we chose to use only extracts in subsequent trials in order to compare samples collected at different times that were extracted for stable storage, and to limit our inference to differences in photodegradation kinetics arising from DOM composition and not matrix differences. Other studies will have to choose between using extracts vs. raw water in accordance with study priorities. The differences we demonstrate (Figures 6 and 7, Table 1) illustrate underappreciated limitations to reproducibility and identify opportunities for further research into the effects of extraction on photodegradation kinetics and mechanisms of action (e.g. effect of matrix vs. fraction of DOM extracted). Along similar lines the referee notes "The authors also advise using a standard optical density in the sample, yet also show there is a concentration dependence to the kinetics, even for an optically thin exposure conditions. Again, a decision will have to made of whats more important, knowing how the DOM behaves at its natural concentration or being able to compare it to other sources." Again, we are in agreement with the referee here, and believe one of our manuscript's contributions is highlighting these underappreciated sources of variability in DOM photodegradation kinetics. This issue in particular is vexing, as differences between DOM concentrations under conditions meeting conventional thresholds for optically thin solutions are not explained well by currently published literature. We revised the manuscript text to make it more clear that our goals were not to show only reproducible results but to identify underappreciated sources of variability that need to be controlled, and to use these insights to conduct limited trials with comparable results (lines 71-72; 576-578).

The referee's final point concerns the utility of our approach in a wider sense – "In the end, investigators will need more information to decide whether the additional insight gained from defining photosensitivity kinetics will be worth the efforts needed to perform all these standardization steps. This report documents experimental tests that show what are sources of variability in the procedure but does not show what new understanding is actually being gained from the results (vs what can already be gained from other optical/chemical measurements typically made on DOM)." We agree that our approach is not trivial, and some applications may not really benefit from this level of effort. Our primary research goal was to compare the sensitivity of different DOM sources to photochemical degradation processes at environmentally relevant temporal scales. Measuring kinetics of DOM photodegradation changes seemed the best way to pursue this, but we found existing approaches to describing DOM photodegradation kinetics allowed for possible uncertainties that were barriers to meaningful comparison (as discussed in detail above). We believe the methodological suggestions we present are useful to other researchers in this area pursuing similar goals, and our research findings (seasonal and spatial differences in wetland DOM photo-sensitivity, with implications for understanding DOM composition and the other ecological processes that affect it in this setting) demonstrates the potential utility of this approach and illustrates phenomena worthy of further investigation.

Specific concerns:
*Line 140 "Solar exposure" – This was not solar exposure, but a laboratory set up. Better (?) "Total sample exposure"*
We agree and revised the manuscript accordingly (now line 154).

*Figure 7 and Table 1 – The captions for these describe kL and kSL with units of mol photons m-2. Those are the units of P, the units of k should be the inverse, $m^2$ [mol photons]$^{-1}$.*
We agree and revised the manuscript accordingly (Figure 7, Figure 10, Figure 12, Table A1, Table A2, Table A3 captions).

*Figure 7 and Table 1 apparently show the same data – and both state the statistics of the comparison. Questionable whether both are necessary. Figure 7 caption, however, says see Fig. 5 for data. It appears that Fig. 6 has the data.*
The referee is correct that Figure 7 caption should refer to Figure 6 instead of Figure 5, and also correct that both Figure 7 and Table 1 show the same data. Readers may find the graphical comparison easier to quickly parse, but the tabular presentation may be more useful to future studies seeking comparison. We think the graphical presentation is more valuable to readers in the main text, and moved Table 1 to an appendix upon revision (now Table A1).

*Line 350-360 – Factors affecting storage of filtered water. Previous text only mentioned preparation of filtrates using 0.7 _M glass fiber filters, which allows passage of a substantial bacterial fraction. However, subsequent guidelines caution against used stored material even filtered to 0.2 _M (some bacteria can even get through those). Please clarify what filtration method was used for the stored sample, but in any case (slow) microbial degradation may be another factor resulting in changes in stored material.*
We inadvertently failed to note an important step in our methods – all samples, whether raw water or solid-phase extracts redissolved in water, were filtered through syringe-mounted 0.2 µm cellulose acetate filters that were pre-rinsed with > 30 mL ultrapure C-free water. GFF filters were used as a pre-filter immediately after samples were returned to the lab, before either short-term storage or solid-phase extraction. Furthermore, since 0.7 µm filters were combusted, their true pore size is probably smaller than 0.7 µm (0.3 µm in Nayar and Chou, 2003) and therefore be comparable 0.2 µm filters (Nayar and Chou, 2003). We apologize for the omission and ensuing confusion and revised the manuscript to clarify (lines 95-97). We also revised our manuscript to clarify the methods used in the storage time experiment (lines 194-200). The referee notes that microbial degradation is still possible even when using a filter of 0.2 µm pore sizes – we agree (see Brailsford et al., 2017; Luef et al., 2015), and believe this supports our recommendation to either use extracts or keep storage time minimal and highlights the need for more research in this area (revisions to lines 118-120). We also believe microbial degradation/contamination is minimal during exposures under the solar simulator using optically thin conditions (e.g. Stubbins et al. 2017, 122), but revised our manuscript to mention the full range of possibilities that must be accounted for in any similar experimental set up (lines 139-144). This would include discussion of between-sample cleaning with dilute NaOH and occasional flushes of sample lines with isopropanol (followed by extensive water flushes and testing for DOC residue) to prevent microbial contamination of flow lines.

*Figure 10 – Caption says see Figure 6 for data, data appears to be in Fig. 9. This correction also applies to the caption for Figure A1. Figure 10 could be misleading to the reader, as a quick inspection may suggest that the spread of the multiple points for a particular sample source (and standard deviation in the table) are indications of reproducibility. Actually, only the SRNOM and three of the small wetland points are repeat determinations on the same sample. Separating the points for different sample dates would avoid this confusion, especially given that*

*later on (section 3.3.2) the authors attribute some of the difference in the wetland results with time to possible differences in sample composition and/or previous solar exposure.*

The referee is correct that the captions for Figure 10 and A1 should read "see Fig. 6 for data". We agree that the representation of data in Figure 10 may be confusing. This criticism may also apply to Table 2 (in original submission), where group means and standard deviations for photodegradations of DOM sources with more than one trial are grouped differently – SRNOM represents three actual replicate trials using the same sample source, the three "large wetland" trials represent different samples (collected at same site but at different times), and the five "small wetland" trials represent three replicate trials using the same source and two trials from different sample sources (collected at the same site but at different times). We originally thought it was useful to make the comparison in Figure 10 in this way because of the widespread interest in differentiating DOM sources by setting. By showing fitted parameters as independent data points we thought readers could parse the different types of grouping by reading the text carefully. However, we agree with the referee's concerns – at the very least the sampling dates should be differentiated by using different symbols in the plot (revised Figure 10 and Figure A1). Upon reflection we also believe Table 2 (in original submission) should be changed along these lines – our original intention was to facilitate comparisons between environmental settings, but calculating group means with these data is not appropriate given their different structures. Means for replicated trials are sensible, and it may even be valuable to see how un-replicated trials with samples from different seasons at a site differ, as solely spatial comparisons often do not account for temporal/seasonal variation but readers may still be interested in summarizing data by site. But calculating one mean from data that includes 3 replicated trials of the same sample source and two further distinct samples does not convey useful information. We think publishing tabular data may be useful for future research, but this data would be better represented separately. For size reasons, such a table may be better suited to an appendix. We re-calculated means where appropriate for replicates and listed parameter fits in Tables A2 and A3.

*Line 456ff, section 3.3.1 Here the relative abundances of kinetic components are considered as metrics of DOM composition or other influences on photosensitivity. A problem with the comparative use of fL and fSL is that the two parameters are not independent. Assuming a good fit to the actual time course the two fractions will always sum to approximately 1, which means that any variation in one fraction will be reflected in a complementary variation in the other. Thus, changes in, e.g. fL, with sample date or location (as later discussed in 3.3.2) could be caused by change in the concentration of labile or semi-labile fraction, or both. If the focus is on the rate parameters, this is not so important, but if the magnitudes of the fractions are to be comparative metrics, the need to be independent of each other. For example, ft could be the actual component score, score normalized to t0 total fluorescence, or score normalized to DOC.*

We see the referee's point that fSL and fL are not truly independent. They were fitted as separate coefficients in the nonlinear fitting algorithm applied to the data, but should indeed sum to 1, so either could be expressed as the difference between 1 and its counterpart (i.e. fSL = 1- fL or vice versa). We expressed them separately here in accordance with their separate representation in the equation conventionally used to define the decay model (Equation 2, which Equation 3 then followed), and because they were fitted separately (which would make the choice of which to use to find its counterpart arbitrary). We agree that this causes ambiguities when trying to understand differences in these values between samples, as in the example offered by the referee. We do believe using these values may be informative, as a major part of our claim to novelty rests on pointing out possible connections between these values and environmental phenomena (such as previous exposure to photodegradation or differences in source material composition) instead of basing inference solely on k parameters. We are not sure how to best resolve this issue. ft already represents PARAFAC component fluorescence normalized to time 0 for each sample. Further processing the data to derive a single value (like the ratio of fSL and fL) adds another layer of abstraction to a situation that already involves multiple modeling steps, and seems harder to interpret in a way that materially connects to the phenomena described. The revised manuscript discusses these points (lines 244-247). However, we still think it is valuable to compare fL between samples – even if differences in this value represent multiple possible causes as described by the referee, differences here represent real observed differences in photodegradation behavior. Normalization to [DOC] or to absorbance at some reference wavelength may be useful in this context. We believe this is an area ripe for future research, and view our contribution in part as pointing out the need for attention to previously unexplored dimensions of current paradigms of DOM photodegradation.

*The authors offer the interpretation that the fractions correspond to "pools" of more or less reactive DOM. However, Murphy et al. (2018) interpret biexponential kinetics as possibly reflecting multiple photoreaction pathways, a fast one involving ROS and slower one related to direct photochemistry. How do the authors reconcile these differing interpretations?*
We appreciate the referee's thoughtful reading of our conclusions. Our conceptualization – differences in photodegradation kinetics arising from different relative quantities of "pools" reacting at relatively fast and slow rates – may be unnecessarily general or vague. However, we think the explanation from Murphy et al. (2018) invoked by the referee is not necessarily opposed to the "pools" we ascribe to the model. We consider these "pools" in a general sense. Given the immense complexity of natural DOM, it is unlikely there are two distinct, non-overlapping compositional types of photo-sensitive DOM. Yet the excellent fit of biexponential models across diverse DOM sources requires some explanation. What we refer to generally as reactive pools could more specifically be the pools of DOM subject to the respective reaction pathways suggested by Murphy et al. and their associated rate constants. Or perhaps it is more appropriate to consider a single "pool" with differing capacities for two classes of reactions. We are not able to definitively conclude what lies behind the biexponential model – but believe our novel attention to the f parameters in the model equation represents the kind of exploration that is needed to make progress in this direction. Our revised manuscript discusses these possibilities and clarifies our use of "pools" (lines 284-285; 497-502)

[revised manuscript text omitted]

---

## Author Response (AR2)

**General comments – response to associate editor and reviewers**

We appreciate the thoughtful feedback from the associate editor and reviewers. We changed the text of the Introduction and Abstract to better reflect the novelty of our research and state our research goals more clearly. We also reorganized parts of our results and discussion to connect more clearly and logically with these research goals.

**Response to Reviewer 1**

Overall comments:

> In "Reproducible determination of dissolved organic matter photosensitivity", Armstrong et al. present a new methodology for obtaining reproducible results for photodegradation experiments of DOM SPE extracts. In the end, I believe that this paper is likely publishable, but I am recommending further revision. At its heart, this paper is a methods paper, and it is unclear to me whether the method is superior to less arduous methodologies. Comparisons of this technique to alternative approaches that clearly show why this approach should/could be adapted or clearer demonstrations of how inferences cannot be obtained without this technique are lacking or not (yet) convincingly explained. Below, I elaborate in more detail.

We agree that the approach presented here is more arduous than other methods used to study DOM photochemistry, but hope our revisions illuminate the fruits of and need for this effort. As noted the introduction has been revised to clarify our goals and the relevance of our results. We presented evidence of several previously unreported sensitivities that may affect experimental photodegradation results, which has relevance for the broader research community even if they do not adopt our approach to address these issues. Our research goals included inferences about compositionally similar DOM (the comparison of two wetlands) – we needed a method that was sensitive enough to reliably discern small differences in photosensitivity that were potentially important to the biogeochemistry and ecology of the larger ecological system. In other cases this rigor may not be required. However, both the findings from our specific study system and the broader methodological insights we present are relevant to researchers interested in DOM photochemistry, even if they do not wish to replicate our particular approach.

> In Fig. 6, the authors show that the SPE DOM and whole water samples differed in apparent composition. I would have liked to have seen more discussion for how this would affect ecological inferences. Does the apparent enhanced reproducibility warrant removal of the sample matrix (and loss of environmental conditions/relevance)?

The reviewers asked whether our focus on solid phase extracts to isolate DOM from its matrix would weaken ecological inferences. As the reviewer noted, we employed extracts to improve reproducibility and comparisons between samples (especially if samples required storage before experiments were possible) but we also used extracts because we think more work needs to be done to disentangle the roles of matrix composition and DOM composition in DOM

photochemistry. Our results here in fact emphasize the need for complementary research on potential matrix influences and their relative contributions to DOM photochemistry. Our revised manuscript should make this distinction more clearly.

> Also, did the authors compare the various kinetic parameterizations for the 2 user comparison in Fig. 5… and is the affect of the different users on the modeling greater than that of the sample prep?

The biexponential model parameters for the 2 user comparison shown in Fig. 5 were compared with 2-tailed t-tests that did not indicate differences in mean parameter values. While not conclusive given n=3 for each operator, the same test was able to distinguish between other comparisons using n=3 on either side such as the RO vs PPL test of SRNOM, or wetland DOM PPL vs. SRNOM PPL. We interpreted this as evidence that with proper control of experimental procedures and conditions, variability due to user differences was less than variability from real compositional differences, evidence of method reproducibility not always reported explicitly. We recommend researchers collecting photodegradation kinetics data undertake similar tests of their procedures.

> Is the SRNOM PPL data in Fig. 6 the same as presented in Fig. 5?

They are separate comparisons with different data sets. Initial tests to compare operator effects (as in Figure 5) produced discrepancies. A component fault arising during these initial comparisons may have been responsible, but we also rewrote operating procedures to ensure they left no room for seemingly trivial or arbitrary differences in execution. Then the two researchers took turns collecting the data shown in this figure. The data in Figure 6 was collected earlier by one of the researchers alone, following the same procedures as those later codified in the more exacting protocol. We grouped the data to reflect the comparisons we intended to make before executing the experiments.

> It's not clear to me whether these conclusions require the approach used here. I would have liked to have seen a comparison to the alternative (some combination of fewer datapoints, no manipulation of pH, whole water, etc. that is currently being used in the literature) to demonstrate the superiority of this method or how using an alternate method gives misleading information. (Alternatively, I missed the explanation, and the manuscript would benefit from a clearer statement of methodological benefits.)

We agree that the specific utility of our approach could be brought into greater relief with comparisons to other methods. This is most clear in (revised) lines 557-575 and Fig. 12, where we compared results from our approach to use of absorbance data alone to highlight what is possible with the fluorescence kinetics. There is also more discussion of our focus on isolating influence of DOM composition on photodegradation potential in lines 426-449.

> They then suggest a series of speculative hypotheticals for how photosensitivity differences might have implications for other ecosystem processes. These remain hypotheticals at this stage, but if the method is demonstrated to be superior/necessary,

answers to many of the questions that are posed would be quite valuable.

We thought additional tests to explore the hypotheses generated by our results would be overwhelming in a manuscript that already explores several experiments, though we hope to continue these lines of inquiry in the future.

> In the Introduction, line 70, the authors state that the goal is "to compare the photosensitivity of DOM sources…." But this is misleading. That may be the ultimate goal, but it is not a major focus in this paper which seems to be a proof of concept for the methodology – as noted by the title of the manuscript. I recommend revising the stated goal to reflect the actual work.

We agree and have revised the manuscript to state the scope of our goals more clearly.

> I found the storage comparisons to be useful but distracting from the main body of the manuscript. I recommend they be moved to supplemental information.

We mostly agree. The storage experiments are mentioned in the main text but the figure with results and description of its relevance are now in Appendix B to improve the flow of the paper.

> The term DOM photosensitivity adds to the already confusing dictionary of photochemical jargon. -why not use DOM potential photodegradation which is the definition provided by the authors for photosensitivity?

We disagree. Avoiding unnecessary jargon improves our science but the meaning of "photosensitivity" seems intuitive in the context of audiences who are interested in photochemical properties of DOM and the mechanisms of its chemical transformation when exposed to sunlight. It seems easier to read "photosensitivity" in a sentence than "DOM potential photodegradation" in this context of this manuscript and related works, as long as the meaning of the term is clearly communicated.

> In the paper and in the responses to one of the reviewers, the authors note that their approach may not be suitable for every situation and that researchers will have to make that decision for themselves. But ultimately, the authors are writing this paper because they hope the method will be adopted in the future. I suggest the authors make specific recommendations for when using this approach makes sense (and maybe suggest when it does not).

We have expanded our thoughts on this point in lines 557-589, though did not offer any ironclad rules or guidance. Ultimately it is up to the researcher to decide which methods are best suited to their questions, and those criteria will vary between individuals.

> "optically thin" is mentioned in line 135 but not defined until line 315. Move its definition to first mention.

We changed the wording of the first occurrence to focus on the purpose of using optically thin samples to save the term and its definition for a more substantial discussion of the concept relevant to our results.

> line 172, are there any effects of adding Cl with HCl? Removing halides was mentioned as a benefit of the PPL step.

This is an excellent question. For some applications this could create problems – for example, investigations into disinfection byproducts. However, pH control was an overriding concern and alternative acids all have their own drawbacks – for example nitric acid absorbs in UV ranges (potentially affecting both photodegradation itself and optical measurements), formic acid would add organic C and absorbs in the UV range. We decided HCl additions would be the least disruptive, most feasible approach.

> -line 290 – it would be beneficial to state what the two types of reactions are.

The revised manuscript includes more information in lines 495-499 and 519-523.

> In Fig. 4, the results of the stir/no stir comparison are not discussed. Stirring does not appear to have much effect on the PARAFAC results (less effect than is seen for Fig. 5), is that correct?

This is correct. We compared data from trials with and without a stir bar present in the equilibration flask and the results did not differ. This did not seem as important as other methodological insights but we have added our observations to the text (lines 314-315).

> In Fig. 8 caption, what are DF, FN, and QB?

These are site codes identifying different wetland locations. We agree these designations are not meaningful in this context, distracting the reader, and have modified the caption.

> nitrite actinometry is an important recommendation from this study but is not described in detail in the methods. Please add these details so researchers can replicate your work.

We agree and have expanded our description in the text (lines 114-124).

> line 439, a sentence was started but never finished.

We were not able to find the fragment described here.

**Response to Reviewer 2**

> Line 117: Can the authors be more specific what "longer storage" means? In the cited Murphy et al. (2018) study I could not find anything on the effect storage has on bulk

DOM vs PPL extracted DOM. Can they provide an estimate how long bulk DOM can be stored (I assume frozen) for fluorescence analysis as opposed to PPL extracts?

We appreciate the reviewer's careful attention to our clumsy word choice in reference to this citation. This question deserves more thorough investigation than can be presented here. In our samples, optical properties of DOC-rich wetland water (20-40 mg DOC L$^{-1}$) were not stable after only 2-3 weeks while stored at 4°C after filtration to 0.2 μm. However, one of the authors of the current manuscript has observed apparent stability in DOC-poor seawater sample optical properties after storage at 4°C for up to a year. We avoided freezing samples to prevent unwanted flocculation of DOM and any possible subsequent alteration of chemical composition as it left and re-entered solution during freezing and thawing. We recommend storage decisions be informed by direct experimentation and reference to published literature on DOM with similar expected composition, but DOM from different settings may have different requirements in this regard.

Line 119: If the samples are frozen, microbial activity should not play role. How do the authors store the samples prior to analysis?

As noted above the samples were not frozen to prevent possible flocculation, and were instead stored at 4°C, so preventing direct microbial activity required filtration. Extracellular enzyme activity may still be possible in these conditions but we assumed it would be minimal, and available substrates quickly exhausted without replenishment by the microbial community.

Line 146: It would be nice to see a picture of this set-up or some schematic drawing. This information could be provided in the suppl. material.

We agree, a photograph is now included in Appendix A (fig A1).

Line 166: The authors explained in the response to the reviewers why this pH was chosen. Nevertheless, I would appreciate a comment here how a pH of 3 corresponds to the expected pH of the samples that were analyzed in this study.

The pH was not chosen to represent the expected pH of similar DOM in environmental settings, but to prevent pH from changing over the course of the experiments themselves, which interferes with the fluorescence kinetics measurements that are the heart of our data. We explain our reasoning in lines 130-135.

Fig. 1 I would mention in the figure caption that solid lines represent emission spectra and dotted lines excitation spectra.

We appreciate the feedback and have amended the figure caption.

Line 258 onwards: Why aren't the components 3 and 4 referred to respective F520 and F450 from here on in the manuscript? This would make it easier to compare to the

previous Murphy et al. (2018) study.

This is an excellent suggestion but we worried about miscommunicating. While the similarities between our components and those in the Murphy et al. 2018 study are strong (and supported by high TCC scores), they are not equally similar for both our components. Component 4 in particular had a lower excitation spectrum congruence to F450. As there is no clearly established hard threshold for comparing component identities, we thought it was important to be clear that we were using our independently-fitted PARAFAC components, not projecting our data onto the model fitted to one of the data sets in that paper. There is a real need to clarify standards for nomenclature when discussing results from multiple PARAFAC models!

> I find the structure of the Results and discussion a bit confusing. I would have expected the authors to first start with the method evaluation and comparison, i.e. comparing bulk DOM and PPL extracts, etc. instead of directly starting with the photodegradation experiments. Before the results of the experiments can be discussed I would first want to see that the method is robust. Also, from the wide number of experimental set-ups described in the methods, it was hard for me to directly understand which experiment the authors are discussing first. The results and discussion would benefit from a more structured presentation. If they want to keep the current order an introductory paragraph explaining what will be described in the following sections would be helpful.

As noted above, we agree and have changed the structure of the introduction and results to improve the clarity of our work.

> Line 281: why is Fig. 10 cited here, when next in line of figures should be Fig. 3?

This was confusing but should be remedied with the revised manuscript structure.

Section 3.1.4: Can the authors refer to how their results relate to other studies that compare PPL and bulk DOM extracts?

> Fig. 7: Please define what is meant with SL (= semi labile) and L (= labile), it is hard to find this information in the text and it would thus be helpful if this abbreviation is defined in the figure caption.

These definitions were presented most explicitly in a much earlier paragraph, in line with some of the other structural issues that made the manuscript's organization confusing, so it should now be easier to refer to this in the text. The definitions have also been added to the figure caption.

> Fig. 8: Please clarify what DF, FN and QB are referring to. This figure only shows the time series on the whole water samples, did the authors also do this experiment for the PPL samples? Do they change if they are stored over longer periods of time?

Noted above – these are site codes identifying different wetland locations. We agree these designations are not meaningful in this context, distracting the reader, and have modified the caption. The storage experiment was only performed for whole water samples, as longer storage periods of solid phase extract methanol eluate at -20°C has been shown to produce consistent sample composition (1H NMR) before and after storage in another study, and effects of storage of methanol eluate at -20°C on DOM chemical composition was thoroughly investigated by Flerus et al (2011) and found to be acceptably stable using concentrated seawater DOM.

Lines 431 onwards: Mention and evaluate again in this paragraph the potential for the (quantitative) loss of compounds during PPL extraction and how this may affect the outcome and interpretation of photodegradation experiments. It does seem like this loss of compounds does not affect the PARAFAC model, but do the authors think that this is universal for all different types of samples from different environments or may the loss of compounds have more profound effects on variations in FDOM?

We think this is an area ripe for further research. We found that RO SRNOM and SRNOM PPL photodegradation time series produced indistinguishable PARAFAC models, but the differences in kinetics suggest the extraction is altering the composition of the DOM in some way that subtly affects its photochemistry. The clearest differences in the biexponential model parameters were in kSL for both PARAFAC components modeled. The reverse osmosis treatment (and subsequent clean-up with cation exchange) used to concentrate the SRNOM when it was collected removes some matrix constituents, but sodium is concentrated during the process, and the RO material contains metals and likely contains silica oxides and sulfate. The solid-phase extraction process should remove many of these, so they may have an effect on photodegradation that is reflected in our results. (Green 2014, Kuhn 2014). Our work demonstrates the need to better understand the effects of strictly DOM chemical composition vs. matrix composition influences on DOM photochemistry, and our use of solid phase extracts reflects our desire to parse these phenomena more carefully.